# From CBC to clarity: Interpretable detection of beta-thalassemia carriers in imbalanced datasets

**Saim Chishti**[1,2], **Faryal Nosheen**[2], **Joddat Fatima**[1,3], **Nadia Sultan**[1,4*], **Madiha Khalid**[5]

1 Center of Excellence in AI (CoE-AI), Bahria University, Islamabad, Pakistan, 2 Department of Computer Science, Bahria University, Islamabad, Pakistan, 3 Department of Software Engineering, Bahria University, Islamabad, Pakistan, 4 Department of Electrical Engineering, Bahria University, Islamabad, Pakistan, 5 Department of Computing, National University of Sciences and Technology (NUST), Islamabad, Pakistan

* nadiaimran.buic@bahria.edu.pk

## Abstract

Thalassemia is an inherited blood disorder and is among the five most prevalent birth-related complications, especially in Southeast Asia. Thalassemia is classified into two main types—alpha-thalassemia and beta-thalassemia—based on the reduced or absent production of the corresponding globin chains. Over the past couple of decades, researchers have increasingly focused on the application of machine learning algorithms to medical data for identifying hidden patterns to assist in the prediction and classification of diseases and patients. To effectively analyze more complex medical data, more robust machine learning models have been developed to address various health issues. Many researchers have employed different artificial intelligence-based algorithms, i.e., Random Forest, Decision Tree, Support Vector Machine, ensemble-based classifiers, and deep neural networks to accurately detect carriers of beta-thalassemia by training on both diseased and normal test reports. While genetic testing is required by doctors for the most accurate diagnosis, a simple Complete Blood Count (CBC) report can be used to estimate the likelihood of being a beta-thalassemia carrier. Various models have successfully identified beta-thalassemia carriers using CBC data alone, but these models perform classification and prediction based on normalized data. They achieve high accuracy but at the cost of substantial changes to the dataset through class normalization. In this research, we have proposed a Dominance-based Rough Set Approach model to classify patients without balancing the classes (Normal, Abnormal), and the model achieved good performance (91% accuracy). In terms of generalization, the proposed model obtained 89% accuracy on unseen data, comparable to or better than existing approaches.

## 1. Introduction

Beta-thalassemia (β-thalassemia) is a genetic disorder that may remain asymptomatic in many individuals, who unknowingly act as carriers and transmit the condition

**Data availability statement:** The data underlying the results presented in the study comprises sensitive medical records and anonymized CBC reports. However, in order to support research transparency and replication, interested researchers may directly contact the Director of MPL for data access requests, subject to approval by MPL and signing of a confidentiality agreement. The contact details are as follows: Dr. Najam Farooq Director Laboratory Metropole Laboratories Private Limited (MPL) Email: najam@fsgroup.pk Website: https://mpl-labs.pk.

**Funding:** The author(s) received no specific funding for this work.

**Competing interests:** The authors have declared that no competing interests exist.

to future generations. This disorder is caused by mutations or alterations in the beta-globin gene (HBB), which is located on chromosome 11. Nearly 200 mutations have been identified as potential causes of β-thalassemia. While gene deletions can also lead to this disorder, they are relatively rare [1]. β-thalassemia presents a significant public health burden in various parts of the world. The global carrier frequency of β-thalassemia is approximately 1.5%, translating to approximately 80–90 million carriers worldwide [2]. Mediterranean regions such as Sardinia and Cyprus report notably high frequencies, approximately 12% to 15% [2]. Southeast Asia has one of the highest carrier frequencies, ranging from 5% to over 10% in various populations [2,3]. Each year, about 60,000 children are born with β-thalassemia major in developing countries [3]. Pakistan, with an estimated population of over 225 million, is believed to have around 10 million β-thalassemia carriers, representing 5% to 7% of the total population [4]. Approximately 5,000 children are diagnosed with β-thalassemia major annually in the country [4], placing a considerable burden on healthcare infrastructure. Early and accurate diagnosis could prevent many of these cases and save lives. Therefore, we focused on local data collection to accurately measure the danger of β-thalassemia carrier presence in terms of local demographics.

Since β-thalassemia carriers are generally asymptomatic, cost-effective screening is essential. A Complete Blood Count (CBC) test, which is rapid and inexpensive, is commonly used to detect carriers. Additional diagnostic tests—such as serum iron, HbA2 levels, ferritin, and iron-binding capacity—can also aid in diagnosis but are more expensive and not universally available in all healthcare settings.

The Mentzer Index (MI) is a very useful method for differentiating between iron deficiency anemia and β-thalassemia trait. The Mentzer Index is calculated by simply dividing Mean Corpuscular Volume (MCV) by Red Blood Cell Count (RBC). If the Mentzer Index is greater than 13, it means the patient has iron deficiency anemia and if the value is less than 13, it means that the patient has beta thalassemia trait [5]. This technique is particularly useful as a first-line screening method in areas where both iron deficiency and β-thalassemia are common [6]. Besides the Mentzer Index, several other hematological indices have been proposed to identify β-thalassemia carriers based on CBC results. When integrated with machine learning techniques, such data can significantly enhance the identification of thalassemia carriers, making it highly suitable for cost-effective screening in resource-constrained settings. When combined with machine learning methods, such data can aid in identifying thalassemia carriers therefore, making it highly suitable for cost-effective screening in resource-constrained settings [7].

In our current study, we applied the Dominance-based Rough Set Approach (DRSA) on a local dataset. DRSA is an extension of Classical Rough Set Theory (RST) that focuses on the dominance aspect of attributes and defines dominance relations [8]. We use the DRSA for calculating the reduced features, then used the Domlem algorithm (Variable Consistency Model) to generate decision rules. Domlem is a rule-induction algorithm that focuses on generating decision rules by considering the dominance relationships between different attributes [9,10]. These decision rules have been used to build a classifier for the dataset. The novelty of this research lies

in the application of DRSA to an imbalanced medical dataset. Additionally, we introduced and evaluated a new index, Beta Thalassemia Detector (BTD), alongside traditional indices, to enhance predictive performance in this specific clinical context.

In the remaining paper, the following sections have been included: In Section 2, a brief definition of the basic concepts of DRSA and the Domlem algorithm. In Section 3, a review of related work is included. Research gaps are based on the literature review outlined in Section 4. In Section 5, details of the proposed methodology are presented. Model evaluation and results are discussed in Section 6. In Section 7, the conclusion and future work of this paper is given.

## 2. Basic concepts of DRSA and Domlem algorithm

In this paper, we have employed the Dominance-Based Rough Set Approach (DRSA). In this section, we briefly present some core concepts of DRSA.

A decision system is a representation of a dataset that consists of a finite set of objects with conditional and decision attributes. In DRSA, it is represented as:

$$\alpha = (U, Q, V, F)$$

where $U$ is the universe, $Q$ is a set of conditional and decision attributes, $V$ consists of possible values that attributes can have, and $F$ defines a dominance function. In DRSA, the core concept is the **dominance relation**, which defines how one object compares to another based on multiple attributes. An object $x$ is said to dominate another object $y$ if $x$ is at least as good as $y$ across all considered attributes. It is named a dominance positive relation:

$$D_P^+(x) = \{y \in U; x D_p y\} \tag{1}$$

If object $x$ is dominated by another object $y$ if object $y$ is more preferred that object $x$ across all considered attributes, it is termed as a dominance negative relation of x:

$$D_P^-(x) = \{y \in U; y D_p x\} \tag{2}$$

The dominance relation is perfect for handling ranked or ordinal data because it encapsulates the concept of "better than" or "worse than "DRSA is beneficial when attributes are preference-ordered. For instance, in a credit evaluation scenario, the decision attribute includes "low credit risk," "medium credit risk," and "high credit risk," reflecting a clear preference order. DRSA integrates this preference structure into the decision rule induction process, leading to more meaningful decision rules. In DRSA, the concepts of lower and upper approximations are adapted from classical rough set theory to work with dominance relations.

### 2.1. P-lower approximation

The **P-lower approximation** of a decision class $C_i$ consists of all objects that can be confidently classified into that class based on dominance. An object is included in the lower approximation if it dominates all other objects in the class (for upward unions) or if it is dominated by all objects in the class (for downward unions).

Mathematically, the P-lower approximation is defined as:

$$\underline{P}\left(Cl_t^{\leq}\right) = \left\{x \in U : D_P^-(x) \subseteq Cl_t^{\leq}\right\} \tag{3}$$

This means that all objects in the lower approximation either dominate or are dominated based on the decision class in question.

## 2.2. P-upper approximation

The **P-upper approximation** of a decision class includes all objects that could potentially belong to that class, meaning they are either in the class or could be in the class based on dominance. An object is included in the upper approximation if it is not dominated by any object outside the class (for upward unions) or does not dominate any object outside the class (for downward unions).

The P-upper approximation is defined as:

$$\overline{P}\left(Cl_t^{\geq}\right) = \left\{x \in U : D_P^-(x) \cap Cl_t^{\geq} \neq \varnothing\right\} \tag{4}$$

This means that the upper approximation includes objects that might be classified into the decision class based on dominance.

## 2.3. VC-Domlem algorithm

The Domlem algorithm, designed for rule induction using lower approximation sets of the Dominance-based Rough Set Approach (DRSA) [9], is inspired by the LEM2 and Modlem algorithms. Domlem generates decision rules based on lower approximations of the upward and downward union of classes. The algorithm focuses on inducing minimal decision rules by considering the preference order of decision classes, starting from the more preferred class for upward unions and the least preferred for downward unions. The process iteratively selects the best elementary conditions for rule creation using a specific evaluation function. This evaluation is based on the ratio of objects covered by a condition compared to the total number of objects, prioritizing the highest ratio. The algorithm aims to generate rules that cover objects efficiently while testing minimalists, focusing on intersections of class approximations and using conditions based on "lower" and "upper edge" objects. Domlem's complexity is polynomial, and it works through a series of steps to induce minimal robust decision rules while handling inconsistencies in dominance relations. An illustrative example demonstrates how Domlem iteratively constructs rules by evaluating conditions, covering objects, and refining rules until a minimal set is obtained. To address this, the VC-Domlem (Variable Consistency Domlem) algorithm was introduced as an extension [10]. VC-Domlem relaxes the strict consistency requirement by incorporating a variable consistency threshold, allowing for a controlled level of inconsistency when inducing rules.

## 3. Related work

In the medical domain, experts predict thalassemia carrier based on blood tests and advanced genetic testing. It may cause delays in prediction and advanced testing is considered expensive for the majority of the population of developing countries. These constraints can hinder proper control of this genetic disorder which is predominantly affecting Asian countries. Therefore, many researchers were attracted to this domain and implemented different algorithms, i.e., Random Forest, Naïve Bayes, KNN, SVM, ANN, etc., to assist in the prediction of different types of thalassemia.

Mo et al. [11] propose a deep neural network (DNN) model based on red blood cell (RBC) indices to enhance thalassemia screening, outperforming traditional statistical methods. Utilizing a dataset of 8,693 records from genetic and CBC tests, the model achieved significant improvements across metrics, notably in accuracy and specificity. While effective, the model's complexity, use of normalization techniques for balancing the data, and lack of interpretability may hinder clinical adoption. In this study [12], Uçucu and Azik use machine learning, specifically artificial neural networks (ANN) and decision trees, to differentiate between β-thalassemia minor (BTM) and iron deficiency anemia (IDA) using only CBC data. The dataset comprises 396 individuals, with 216 IDA and 180 BTM cases, collected from Muğla Sıtkı Koçman University Training and Research Hospital between January 2015 and June 2021. Hematological parameters and ferritin levels were analyzed. The artificial neural network (ANN) model achieved 99.5% accuracy, outperforming traditional

indices (e.g., Green & King), though a small dataset may cause overfitting which lacks generalization. Rane et al. [13] reviewed various machine learning(ML) algorithms, including Logistic Regression, Naive Bayes, K-Nearest Neighbor, Support Vector Machine, Decision Tree, and Random Forest, evaluating their effectiveness in anemia detection and classification. The study finds that algorithms like Random Forest and Decision Tree frequently outperform others in accuracy for anemia prediction. The dataset utilized in this research includes comprehensive health-related data such as hemoglobin levels, socio-economic status, and demographic factors, with data sourced from medical institutions, public health surveys, and databases like the NFHS-4. Despite the dataset's variety, the study notes the challenges posed by class imbalances and the need for larger, balanced datasets to improve model generalizability and robustness in clinical applications. Ali and Saqib [14] employed data mining classifiers, including J48 Decision Tree, Naïve Bayes, SMO (Sequential minimal optimization), and Multilayer Perceptron, to predict thalassemia risk, achieving high accuracy, with Naïve Bayes yielding the best results at 99%. This work leverages a dataset of 301 CBC reports from the National Institute of Blood Diseases (NIBD) in Karachi, which includes variables such as gender, CBC blood report parameters indicating positive or negative thalassemia statuses. Although effective, the study's limitations include a relatively small dataset size and reliance on CBC attributes, which may miss other factors influencing thalassemia prediction accuracy. Pullakhandam and McRoy [15] developed machine learning models, including Gradient Boost, to classify iron deficiency anemia (IDA) using Complete Blood Count (CBC) data alone, eliminating the need for expensive serum ferritin tests. The dataset, sourced from the NHANES (National Health and Nutrition Examination) survey, included over 19,000 instances, along with a smaller validation dataset from Kenya, enabling robust cross-validation. Key features, such as hemoglobin levels, age, red blood cell distribution width, and pregnancy status, were crucial for classification, achieving a Precision-Recall AUC of 0.87 and a sensitivity of 98% on the NHANES data. While effective, the study's limitations include class imbalance (minority class) and the lack of ferritin level adjustments for inflammation, which may affect the accuracy in clinical applications. Moreover, it does not further identify patients as thalassemia carrier or not. Ferih et al. [16] examined the application of artificial intelligence (AI) in diagnosing and managing thalassemia, focusing on algorithms designed to differentiate thalassemia from other forms of microcytic anemia, particularly iron deficiency anemia. The study reviewed multiple models, including k-nearest neighbor, random forest, support vector machine, and artificial neural networks. These models are used to predict beta-thalassemia or differentiate beta-thalassemia and iron deficiency. Although these models achieved high sensitivity and specificity but do not address the concerns related to imbalance datasets. Moreover, the majority of methods reliance on black box prediction techniques, which may restrict the generalizability of results across diverse clinical settings. Saleem et al. [17] conducted a comparative analysis of feature selection techniques for thalassemia prediction, utilizing classifiers like K-Nearest Neighbors, Decision Trees, Gradient Boosting, Linear Regression, and Support Vector Machine. The dataset comprised 616 thalassemia patient records from Kaggle, which consisted of CBC parameters such as Hemoglobin concentration, Mean Corpuscular Volume (MCV), Hematocrit (Hct), Mean corpuscular hemoglobin concentration (MCHC), Mean corpuscular hemoglobin (MCH), Red blood cell (RBC) count, and red cell distribution (RDW). Another dataset with 203 records provided additional CBC indices and gender details. The study found that Gradient Boosting achieved the highest accuracy, but a small dataset and the lack of extensive feature diversity may limit generalizability across larger clinical populations. Long and Bai [18] constructed a machine-learning model to predict thalassemia in pregnancy using blood routine indicators, aiming to reduce the cost and time associated with genetic testing. The dataset included 7,621 cases from pregnant women screened for thalassemia in Chongqing, China, from 2018 to 2022. Key variables, such as MCV, MCH, RBC, and MCHC, were identified through Least Absolute Shrinkage and Selection Operator (LASSO) regression and were found to be highly correlated with thalassemia. The predictive model achieved an AUC of 0.911, demonstrating strong effectiveness. Limitations include a focus on common thalassemia genotypes, which may restrict the model's applicability in populations with varied genetic backgrounds. Ali and Abdulazeez [19] examined recent advancements in machine learning (ML) applications for diagnosing thalassemia, highlighting the impact of various algorithms, such as predictive modeling,

image analysis, and deep learning, on improving diagnostic accuracy. The review spans multiple ML techniques used across diverse biomedical datasets, including genetic and CBC data, to assess diagnostic effectiveness. Limitations identified in the study include challenges related to data diversity, model transparency, and the need for robust and varied datasets, which are essential for enhancing model generalization across different populations and clinical settings. Obstfeld et al. [20] explored the applications of machine learning (ML) in laboratory hematology, focusing on the automation of tasks such as blood smear analysis, white blood cell classification, and morphological grading. The review highlights the role of ML models, including deep learning algorithms, in improving diagnostic efficiency and reducing manual labor, with examples like Cella Vision systems used in hematology laboratories to classify cell types with high sensitivity and specificity. However, limitations include challenges with standardizing data across laboratories and the potential for model errors when trained on non-diverse datasets, which may limit generalizability and accuracy in varied clinical contexts. Wiratchawa et al. [21] introduced "ThalNet," a deep learning model designed for thalassemia screening and subtype classification using blood image structure-function imaging (ISFI) data. This model uses EfficientNet-B7, enhanced by converting human blood sample images into a differential dynamic image format for analysis. The dataset consists of 268 images for thalassemia versus normal classification and 168 images for distinguishing β-Thalassemia subtypes, derived from pDDM device recordings and CBC parameters. ThalNet achieved an accuracy of 86% for screening and 67% for subtype classification. Limitations include the small dataset size and a dependency on high-quality images, which may impact model performance in diverse clinical settings. Tran et al. [13] investigated the prevalence of thalassemia in the Vietnamese population and developed a clinical decision support system (CDSS) for prenatal screening. The dataset comprised over 10,000 medical records from pregnant women and their husbands, including complete blood count (CBC), high-performance liquid chromatography (HPLC), and iron status test results. Genetic testing was conducted for selected cases, identifying 10.73% alpha-thalassemia and 2.24% β-thalassemia prevalence. The CDSS incorporated both expert and AI-based systems, with the multilayer perceptron (MLP) model achieving 98.5% accuracy, demonstrating the effectiveness of AI in prenatal screening. Limitations include dataset imbalance and reliance on CBC parameters, which may affect generalizability across diverse clinical settings.

## 4. Research gaps

In the above section, we have discussed different approaches related to the identification of β-thalassemia carrier. The majority of these approaches are the successful application of random forest, deep neural network, and other algorithms after applying normalization techniques on the dataset, i.e., Synthetic Minority Oversampling Technique (SMOTE), standard scaling, etc. The research, however, still requires more probing of efficient solutions in case of an imbalance dataset with a minority class and decision/classification must be interpretable to decision makers. In the medical field, dataset consists of records of patients that need to be handled carefully to avoid any inconsistencies or ambiguity that may lead to miss-classification or incorrect prediction. In this paper, we proposed a technique to predict β-thalassemia carrier based on CBC reports only, using an imbalanced dataset. Our proposed model is based on a heuristic rules-based DRSA approach [22] which is computationally efficient in terms of time and memory. Moreover, using lower approximation sets of DRSA, the VC-Domlem algorithm induced if-else rules for classification. These rules are human-interpretable and can be traced back to enhance the trust of doctors and professionals in AI. In the context of an imbalance dataset, our proposed model results in better performance in terms of accuracy.

## 5. Methodology

Here, we discuss in detail the locally acquired dataset, pre-processing procedure, proposed framework for feature reduction, and early detection of β-Thalassemia carrier. We have used dominance-based rough set theory (DRSA) for feature extraction and then formulated rules based on lower approximation sets of DRSA using the VC-Domlem algorithm [15]. For efficient computation, the optimized algorithm has been used to compute DRSA approximations [22]. This approach

is based on the logical properties of DRSA proposed by Greco et al. [8], used heuristic rules to avoid redundant computations. A brief overview of the proposed framework presented in this research is given in Fig 1.

## 5.1. Dataset description

In this study, a local dataset has been collected in collaboration with Metropole Lab Pvt. Ltd., (MPL) Islamabad, Pakistan. While collecting data, privacy guidelines have been strictly followed to ensure a patient's privacy, therefore all data was anonymized. Metropole lab has been conducting molecular genetic testing to identify various genetic disorders; thalassemia is one of them. At MPL, advanced genetic screening is performed to identify β-Thalassemia carrier. In this research, the study has been conducted on blood screening reports of 3816 patients from 14-Aug-2020 to 24-Jun-2024. Prior to transfer, MPL's data-security team automatically stripped patient names, national IDs, contact details, accession barcodes and all other direct identifiers, leaving no code key with the investigators; thus, the dataset is anonymous as defined by the National Bioethics Committee of Pakistan. Because the research involved only secondary analysis of fully anonymized data, the Institutional Review Board confirmed the study as *exempt* and waived the requirement for informed consent. The dataset was accessed for research purposes only once. The anonymized CBC dataset was exported from MPL's laboratory-information system on 15 July 2024 and transferred to the research team on 20 July 2024; all analyses were completed by 31 October 2024. All procedures comply with the Declaration of Helsinki, national privacy regulations, and MPL's internal confidentiality policy.

In this data, 17% of patients are β-Thalassemia carrier and 82.4% are β-Thalassemia non-carrier. The findings of the reports have been diligently verified by the hematologist prior to making the final decision. The dataset comprised patients belonging to different age groups, genders, and regions. The decision attribute of dataset is organized into two classes:

1. The samples with CBC indices belong to normal ranges.

2. The samples with CBC indices fall in abnormal/intermediate ranges. (Further investigated by genetic testing, i.e., electrophoresis, to enhance the reliability of the proposed technique)

The dataset consists of patient CBC reports and has the following parameters:

WBC: White blood cell count

LYMp: A proportion of one type of white blood cell, which is related to immunity of your body, lymphocytes

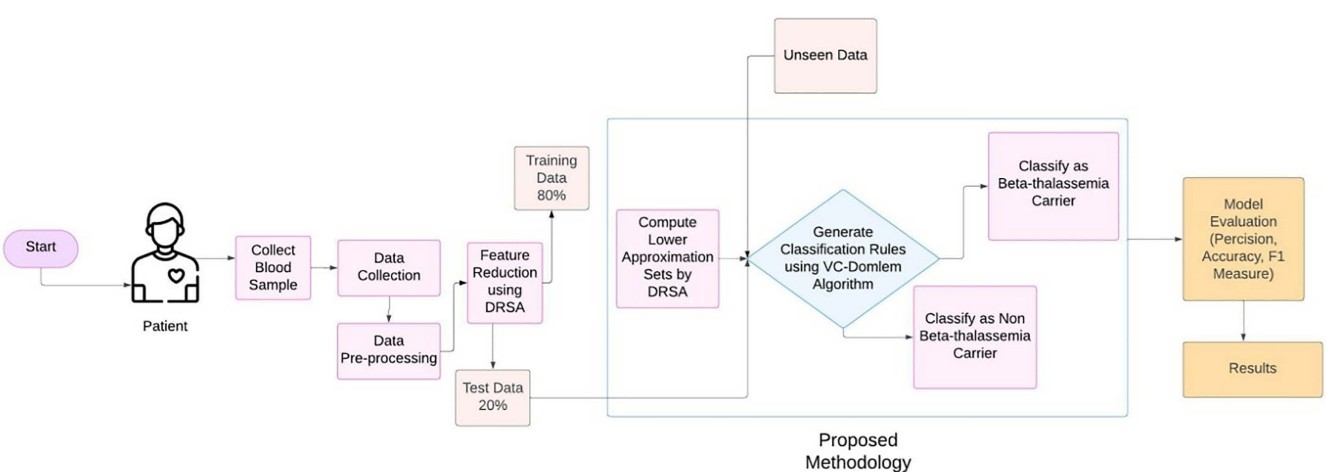

**Fig 1. Proposed Methodology: Early Detection of β-Thalassemia carrier Based on Imbalance Dataset.**

LYMn: Absolute count of Lymphocytes

MIDp: Mid-sized cells, a percentage of different types of white blood cells that is rare.

MIDn: Absolute number of mid-sized cell

NEUTp: a type of white blood cells that is essential to fight infections, and neutrophils.

NEUTn: Absolute count of neutrophils

RBC: Red blood cell count

HGB: hemoglobin, red blood cells which carry oxygen

HCT: Hematocrit, percentage of red blood cells in the blood sample

MCH: Measure the average level of hemoglobin in blood sample.

MCV: Mean Corpuscular Volume, measurement of average red blood cells

MCHC: Distinguish between the red color and pale color of red blood cells by measuring the concentration of Hemoglobin

RDWSD: Measure the shape, size, and width of red blood cells to determine whether all cells are the same.

RDWCV: Measure variability in red blood cell size

PLT: Platelet count, a measure of blood clotting

PCT: Platelecrit, a measure of platelets' volume in blood

PLCR: Ratio of Platelet-Lymphocyte, measure inflammatory

These measures are necessary to find hematological disorders. For thalassemia, researchers suggested some indexes to identify possible thalassemia carrier based on CBC parameters, i.e., MI (Menzer Index), RDWI, GKI, etc. [23]. Details are mentioned in Table 1.

These indexes increase the accuracy of β-Thalassemia carrier detection. In this study, we have evaluated the proposed model by using all these indexes, none of the studied indices provided 100% accuracy but results were improved using these indices in comparison with CBC parameters only. In addition to that, we have evaluated the proposed model using a new index **Beta Thalassemia Detector (BTD)** with MI, which considerably enhances the classification of β-Thalassemia carriers. The formula is given in equation 5.

$$BTD = \left( \frac{RBC}{HGB} \right) \times 100$$

(5)

**Table 1. Formula of discrimination index.**

| Index | Formula | Range for β-Thalassemia Carrier | Range for Non β-Thalassemia Carrier |
|---|---|---|---|
| Mentzer Index (MI) | $\frac{MCV}{RBC}$ | ≤ 13 | > 13 |
| Green and King Index (GKI) | $\frac{MCV^2 \times RDW}{Hb \times 100}$ | ≤ 65 | > 65 |
| MCHD[b] | $\frac{MCH}{MCV}$ | > 0.35 | ≤ 0.35 |
| Red Cell Distribution Width Index (RDWI) | $\frac{MCV \times RDW}{RBC}$ | ≤ 220 | > 220 |
| Ricerca Index (RI) | $\frac{RDW}{RBC}$ | > 4.4 | ≤ 4.4 |

**Ethics Statement:** This research was conducted adhering strictly to ethical standards essential for clinical and medical research. Given that the dataset was obtained from Metropole Laboratories Private Limited (MPL), utmost attention was paid to the confidentiality and anonymity of patient data. The dataset was anonymized to ensure the complete removal of identifiable personal information, maintaining patient privacy and confidentiality, in compliance with the terms outlined by MPL.

**Potential Biases:** The dataset consisted predominantly of regional data from Pakistan, reflecting demographic-specific hematological characteristics. Although this increases relevance locally, it necessitates careful consideration when generalizing findings to broader populations. To mitigate bias due to regional data limitations, in future studies we will incorporate more diverse and geographically dispersed datasets.

**Implications of Misclassification:** Misclassification of β-thalassemia carriers can have serious consequences. False positives may lead to unnecessary emotional distress, financial burden, and further invasive testing for patients, while false negatives might result in the failure to provide timely clinical intervention, perpetuating disease transmission risk. The Dominance-based Rough Set Approach (DRSA) employed in this study enhances transparency and interpretability through rule-based classification, allowing healthcare professionals to clearly understand and validate model predictions, thereby reducing the risk of misclassification.

**Responsible Deployment in Healthcare:** The deployment of this model in healthcare settings should be executed with stringent oversight by medical professionals, with clear guidelines indicating it as an assistive tool rather than a stand-alone diagnostic solution. Continuous model performance monitoring and periodic retraining with updated, diverse patient data are recommended to maintain accuracy and adapt to demographic and epidemiological shifts. Additionally, extensive validation and cross-validation with independent datasets from diverse sources are essential before clinical implementation. Metropole Laboratories, being a government-registered institution, has endorsed this collaborative approach, ensuring domain-expert supervision throughout the development and potential deployment phases.

## 5.2. Imbalance dataset

We conducted a study on an imbalanced dataset where only 17% of patients were β-thalassemia carrier. Mostly machine learning algorithms and deep learning techniques do not perform well when one class is in a minority as compared to other classes of dataset [24,25]. Various methods have been suggested to balance the classes of dataset for accurate classification using ML models but the application of these methods on medical dataset may cause inaccurate classification, prediction, and decisions [24]. The cost of wrong prediction/classification in case of the medical dataset may cost the life of a person [24]. Han et al. [24] suggested that the accuracy of ML models mainly depends on the accurate prediction of diseased people, a minority class.

DRSA is an extension of classical rough set theory, which, along with its various extensions, has been widely utilized by researchers specifically for handling feature selection and classification problems involving imbalanced datasets [26–29]. The theoretical suitability of rough set theory and its variants for imbalanced data lies primarily in their ability to identify significant attribute subsets (reducts) without requiring data normalization or balancing. DRSA's reliance on dominance relationships allows it to effectively capture critical attribute interactions and dependencies, emphasizing minority class characteristics even when they appear infrequently in the dataset. Furthermore, in our research, we specifically employed DRSA on an imbalanced beta-thalassemia dataset to empirically evaluate its performance. To make accurate decisions in case of an imbalance dataset, we have found dominance-based rough set theory as an excellent tool. We also experimented with other techniques, i.e., random forest, SVM, and neural networks but our results demonstrated that DRSA outperforms various other classification algorithms in accurately classifying minority class instances.

## 5.3. Data pre-processing

The data collected from Metropole Laboratories was in the form of Complete Blood Count (CBC) reports of patients. After extracting data from these reports, the initial inspection revealed inconsistencies and missing values, which, if not

addressed, could negatively impact the accuracy of the classification algorithm. To ensure robust and reliable analysis, a structured and clearly defined cleaning process was carried out as follows:

- **Handling Missing Critical Parameters:**

  ◦ Records lacking essential parameters such as Red Blood Cell (RBC) count (3 samples) or Hemoglobin A levels (13 samples) were excluded, given the necessity of these metrics for accurate beta-thalassemia trait identification.

- **Removal of Records with Ambiguous or Missing Remarks**

  ◦ A total of 26 samples with no accompanying remarks were removed as they offered insufficient diagnostic clarity.

  ◦ An additional 3 samples explicitly labeled as "inconclusive" by lab professionals were excluded from further analysis to maintain dataset integrity.

  ◦ Furthermore, 356 samples were ignored due to ambiguous wording within the provided remarks. Specifically, these samples included unclear annotations such as "borderline," "uncertain results," or similarly vague phrasing that could not be confidently assigned to either "Normal" or "Abnormal" categories.

- **Standardization of Remarks:**

  ◦ Samples explicitly labeled as "NOT DETECTED" or "NO ABNORMALITY" were clearly classified into the "Normal" category, aligning with clinical conventions.

- **Duplication and Attribute Reduction**

  ◦ Duplicate entries were systematically identified and removed to avoid biases from repetitive data.

  ◦ Non-essential attributes, including patient age, and gender, were removed to focus exclusively on the CBC parameters relevant to beta-thalassemia detection. However, a supplementary analysis examining the impact of gender revealed negligible influence on model outcomes, validating the decision.

  After the completion of data cleaning and preprocessing steps, the refined dataset depicted in Fig 2 comprised:

**Abnormal samples:** 593

**Normal samples:** 2822

**Total samples:** 3415

### 5.4. Proposed methodology

Dominance-based rough set theory (DRSA) is a supervised algorithm that is based on the set theory of mathematics. In order to properly know the relative importance of various features in decision-making, DRSA integrates the idea of dominance, in contrast to classic rough set theory, which approximates classes based on the dominance relation of attribute values. This method enables more detailed analysis and classification by taking into account which feature values are more preferred as compared to others and affect decision-making. Additionally, measurement errors and differences in clinical methods might lead to inconsistent medical data; DRSA is an excellent tool for handling inconsistent data [8]. Even in the case of imbalance datasets, DRSA is able to successfully classify objects. Many relevant medical attributes such as severity of illness and laboratory test results are ordinal. DRSA effectively manages these ordinal attributes, preserving the natural order of variables. DRSA can handle ambiguity and imprecision, which helps in overcoming difficulties with thalassemia categorization. Reducing features is very important; it improves efficiency by decreasing computation time and memory usage and makes the analysis faster and more effective for larger datasets.

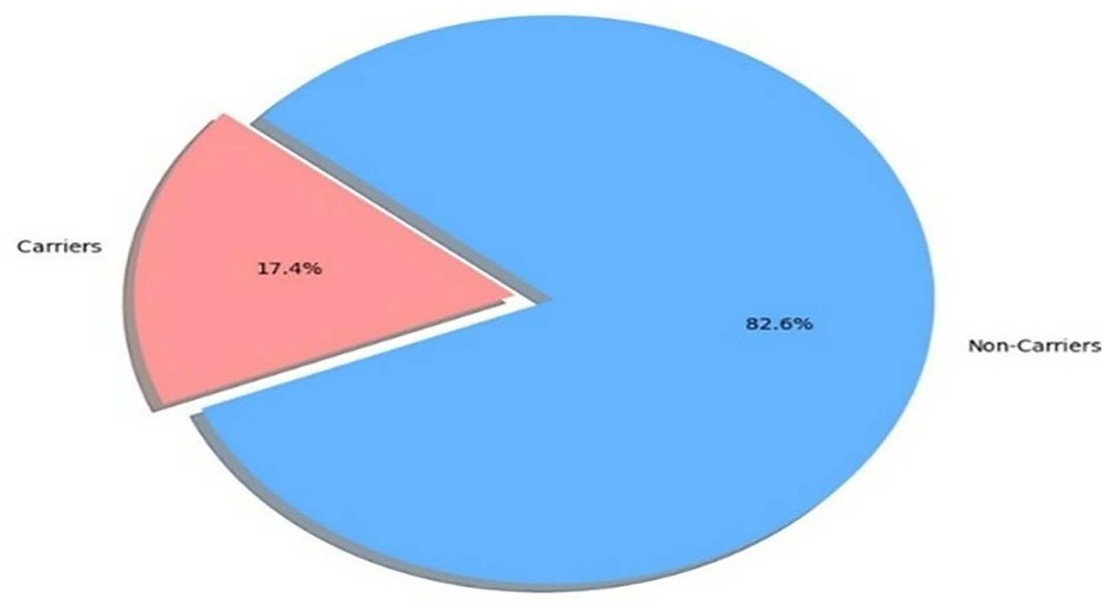

**Distribution of Beta Thalassemia Carriers and Non-Carriers**

Carriers 17.4%

82.6% Non-Carriers

Carriers: 593
Non-Carriers: 2822
Total Patients: 3415

**Fig 2. Dataset CBC reports.**

As mentioned previously, it can be seen that the computations required for calculating lower and upper approximation sets are considered expensive in terms of time and complexity. Heuristic rules proposed in [22], to compute lower and upper approximations are less expensive as compared to the classical DRSA algorithm proposed by Greco et. al. [8].

In this research results highlighted that DRSA algorithm [22] performed well on the imbalanced dataset without requiring any normalization. In the first step, important features were extracted from the dataset using [22] based on the quality of approximations, then classification was performed using decision rules, generated by the VC-Domlem algorithm [9] based on approximation sets of DRSA.

**5.4.1. Feature reduction.** After data pre-processing, the dataset has 19 features (listed in section 5.1). We used a dominance-based rough set approach to perform feature reduction. The DRSA heuristic rules-based algorithm [22] performed feature selection by computing lower and upper approximation sets.

We labeled the decision classes of dataset as:

1. '0' corresponds to class where patients categorized as non β-Thalassemia carrier

2. '1' corresponds to class where patients categorized as β-Thalassemia carrier

As discussed earlier, to compute lower and upper approximation sets, each object needs to be compared with other objects in terms of 19 features. We computed lower and upper approximation sets of upward and downward union of classes for class '0' and class '1' $\{\underline{P}\left(Cl_0^{\leq}\right), \underline{P}\left(Cl_1^{\leq}\right), \underline{P}\left(Cl_0^{\geq}\right), \underline{P}\left(Cl_1^{\geq}\right), \overline{P}\left(Cl_0^{\geq}\right), \overline{P}\left(Cl_1^{\geq}\right), \overline{P}\left(Cl_0^{\leq}\right), \overline{P}\left(Cl_1^{\leq}\right)\}$ and then measured quality of approximations as given in equation (10).

Boundaries are computed by:

$$B_{nP}\left(CI_1^{\geq}\right) = \overline{P}\left(CI_1^{\geq}\right) - \underline{P}\left(CI_1^{\geq}\right) \tag{6}$$

$$B_{nP}\left(CI_1^{\leq}\right) = \overline{P}\left(CI_1^{\leq}\right) - \underline{P}\left(CI_1^{\leq}\right) \tag{7}$$

$$B_{nP}\left(CI_0^{\geq}\right) = \overline{P}\left(CI_0^{\geq}\right) - \underline{P}\left(CI_0^{\geq}\right) \tag{8}$$

$$B_{nP}\left(CI_0^{\leq}\right) = \overline{P}\left(CI_0^{\leq}\right) - \underline{P}\left(CI_0^{\leq}\right) \tag{9}$$

$$\gamma_{P(CI)} = \frac{\left| U - \left( \left( \cup B_{nP}\left(CI_t^{\geq}\right) \right) \cup \left( \cup B_{nP}\left(CI_t^{\leq}\right) \right) \right) \right|}{|U|} \tag{10}$$

where P is subset of complete feature set C and t represents class (0 or 1)

If the quality of approximations for subset P is equal to the quality of approximations computed using a complete feature set ($\gamma_{P(CI)} = \gamma_{C(CI)}$), the set P is called a reduct set [8]. A reduct set is a set of important features which can be used instead of a complete feature set. We computed the quality of approximations for both classes using all 19 features of the dataset and obtained the following results given in Table 2:

We obtained the following reduct sets in Table 3:

We have used Reduct set 2 for generating decision rules to classify patients.

**5.4.1.1. *Illustrative example:*** Table 4 demonstrates a sample of 10 patient records and their CBC-related values, used to illustrate the application of DRSA.

1) **Lower Approximation Sets**

- $\underline{P}\left(CI_0^{\leq}\right) = \{x_1, x_{10}, x_6, x_2, x_4, , x_3, x_8, x_7\}$
- $\underline{P}\left(CI_1^{\leq}\right) = \{x_1, x_{10}, x_9, x_6, x_5, x_2, x_4, x_3, x_8, x_7\}$

**Table 2. Approximations of DRSA.**

| Quality of Approximation: 0.994 | | |
|---|---|---|
| **Class** | **Accuracy** | **Cardinality** |
| For class '0' | 0.993 | 2826 |
| For class '1' | 0.967 | 589 |

**Table 3. Reduct sets.**

| Name | Cardinality | Features |
|---|---|---|
| Reduct set 1 | 5 | HCB, MCHC, RDWI, MI, BTD |
| Reduct set 2 | 5 | HCB, MCHC, RDWCV, MI, BTD |

**Table 4. Dataset of 10 patients (few features have been selected for this example).**

| Patient | RBC Count (mil/cmm) | Hemoglobin Level (g/dL) | Hemato-crit (%) | MCV (fL) | Msquare | MCH (pg) | MCHC (g/dL) | RDW-CV (%) | MI | GK | GreenK-ing | RDWI | Deci-sion |
|---|---|---|---|---|---|---|---|---|---|---|---|---|---|
| x1 | 3.87 | 10.4 | 32 | 83 | 6889 | 27 | 32 | 14 | 21.44703 | 37.21154 | 256350.3 | 300.2584 | 0 |
| x2 | 4.53 | 6.3 | 23 | 51 | 2601 | 14 | 27 | 23 | 11.25828 | 71.90476 | 187024.3 | 258.9404 | 0 |
| x3 | 4.14 | 9.1 | 29 | 71 | 5041 | 22 | 31 | 19 | 17.14976 | 45.49451 | 229337.8 | 325.8454 | 0 |
| x4 | 4.01 | 11 | 34 | 84 | 7056 | 28 | 33 | 17 | 20.94763 | 36.45455 | 257223.3 | 356.1097 | 0 |
| x5 | 3.42 | 7.4 | 24 | 69 | 4761 | 22 | 31 | 17 | 20.17544 | 46.21622 | 220035.4 | 342.9825 | 1 |
| x6 | 4.06 | 10.5 | 32 | 78 | 6084 | 26 | 33 | 17 | 19.21182 | 38.66667 | 235248 | 326.601 | 0 |
| x7 | 4.75 | 10.9 | 34 | 71 | 5041 | 23 | 33 | 16 | 14.94737 | 43.57798 | 219676.6 | 239.1579 | 0 |
| x8 | 4.19 | 5.4 | 23 | 55 | 3025 | 13 | 24 | 24 | 13.12649 | 77.59259 | 234717.6 | 315.0358 | 0 |
| x9 | 5.46 | 10.9 | 34 | 62 | 3844 | 20 | 32 | 16 | 11.35531 | 50.09174 | 192552.7 | 181.685 | 1 |
| x10 | 3.8 | 10.2 | 32 | 83 | 6889 | 27 | 33 | 20 | 21.84211 | 37.2549 | 256649 | 436.8421 | 0 |

- $\underline{P}\left(Cl_0^{\geq}\right) = \{x_1, x_{10}, x_9, x_6, x_5, x_2, x_4, x_3, x_8, x_7\} = Cl_0^{\geq}$
- $\underline{P}\left(Cl_1^{\geq}\right) = \{x_5, x_9\} = Cl_1^{\geq}$

2) **Upper Approximation Sets**

- $\overline{P}\left(Cl_0^{\leq}\right) = \{x_1, x_2, x_4, x_{10}, x_6, x_3, x_8, x_7\}$
- $\overline{P}\left(Cl_1^{\leq}\right) = \{x_1, x_9, x_5, x_{10}, x_2, x_6, x_4, x_3, x_8, x_7\}$
- $\overline{P}\left(Cl_0^{\geq}\right) = \{x_1, x_{10}, x_9, x_6, x_5, x_2, x_4, x_3, x_8, x_7\} = Cl_0^{\geq}$
- $\overline{P}\left(Cl_1^{\geq}\right) = \{x_5, x_9\} = Cl_1^{\geq}$

3) **P-Boundary (P-Doubtful Region)**

The P-boundary region of a class represents the difference between the upper and lower approximations

- $B_{np}\left(Cl_0^{\leq}\right) = \overline{P}\left(Cl_0^{\leq}\right) - \underline{P}\left(Cl_0^{\leq}\right) = \varnothing$
- $B_{np}\left(Cl_1^{\leq}\right) = \overline{P}\left(Cl_1^{\leq}\right) - \underline{P}\left(Cl_1^{\leq}\right) = \varnothing$
- $B_{np}\left(Cl_0^{\geq}\right) = \overline{P}\left(Cl_0^{\geq}\right) - \underline{P}\left(Cl_0^{\geq}\right) = \varnothing$
- $B_{np}\left(Cl_1^{\geq}\right) = \overline{P}\left(Cl_1^{\geq}\right) - \underline{P}\left(Cl_1^{\geq}\right) = \varnothing$

4) **Accuray of Approximations**

The classification accuracy of the DRSA-based lower and upper approximations is summarized in Table 5.

5) **Quality of Approximations**

The quality of approximation measures how well a dataset can be classified based on its attributes:

$$\gamma_p(Cl) = \frac{\left| U - \left(\left(U_{t \in T} B_{np}\left(Cl_t^{\geq}\right)\right) \cup \left(\left(U_{t \in T} B_{np}\left(Cl_t^{\leq}\right)\right)\right)\right)\right|}{|U|}$$

Since there are no objects in the **P-boundary**, the quality of approximation is **1 (100%)**, meaning that the given attributes fully determine the classification of objects.

**Table 5. Accuracy of approximations.**

| Class | Accuracy | Cardinality |
|---|---|---|
| 0 | 1.000 | 8 |
| 1 | 1.000 | 2 |

This example demonstrates how rough set theory is applied to classify patients based on their hematological parameters. Such an approach is valuable in medical diagnosis, where certainty in classification is critical for effective decision-making.

**5.4.2. Classification.** As discussed earlier, DRSA is a highly effective tool where feature values relate to each other and affect decision class. It captures the dominance relationships between the attributes which is crucial for accurate decision making. Moreover, unlike other machine learning algorithms, DRSA efficiently handles imbalanced datasets. Unlike deep learning, which is computationally expensive, and lacks transparency, DRSA handles datasets efficiently while offering clear, interpretable decision rules. This makes it a more practical and trustworthy tool for healthcare professionals.

In this research, using DRSA, we obtained a reduced feature set, which consisted of only 5 features instead of 19 features. It considerably enhanced the efficiency of the classifier. We have used an imbalanced dataset without performing normalization. To design the classifier, heuristic rule based dominance based rough set algorithm [22] have been used. We randomly split the dataset into 80% for training and 20% for testing.

In this first step, we have applied DRSA algorithm [22] and compared each object with other objects in terms of features to compute lower approximation sets using 5 features (HCB, MCHC, RDWCV, MI, BTD). A few objects are listed below (x2918 termed as patient no 2918 and so on):

$$\underline{P}\left(Cl_1^{\geq}\right) = \{x2918, \ x3287, \ x2230, \ x2005, \ x3356, \ x1199, \ x3102, \ x2068, \ x1929, \ x331, \ x2212, \ x1217,$$
$$x364, \ x2264, \ x1757, \ x3406, \ x3236, \ x2706, \ x3008, \ \ldots\ldots, x1248, \ x1609\}$$

$$\underline{P}\left(Cl_1^{\leq}\right) = \{x2918, \ x1820, \ x2230, \ x1514, \ x2568, \ x3383, \ x899, \ x1199, \ x1549, \ x1754, \ x1888, \ x2728,$$
$$x890, \ x2264, \ x2775, x1782, \ x2307, \ x63, \ x308, \ x2964, \ \ldots\ldots\ldots\ldots.., \ x2322, \ x1698\}$$

$$\underline{P}\left(Cl_0^{\geq}\right) = \{x2918, \ x1820, \ x2230, \ x1514, \ x2568, \ x3383, \ x899, \ x1199, \ x1549, \ x1754, \ x1888, \ x2728,$$
$$x890, \ x2264, \ x2775, \ x1782, \ x2307, \ x63, \ x308, \ x2964, \ \ldots., x2322, \ x1698\}$$

$$\underline{P}\left(Cl_0^{\leq}\right) = \{x1820, \ x3383, \ x1514, \ x2568, \ x899, \ x1549, \ x1754, \ x1888, \ x2728, \ x890, \ x2775,$$
$$x1782, \ x2307, \ x63, \ x308, \ x2964, \ x948, \ x1315, \ x3151, \ \ldots\ldots, \ x2157, \ x900, \ x2322, \ x1698\}$$

Based on lower approximation sets, we have applied VC- Domlem algorithm [25] to design if-else rules for classification. These rules successfully classified patients as β-Thalassemia carrier and non β-Thalassemia carrier, in case of an imbalanced dataset. A few rules are listed in Table 6:

# 6. Results and performance evaluation

We have implemented the proposed framework to assist in the early detection of β-Thalassemia carrier and compared our proposed approach for the imbalance dataset with Random Forest and SVM using python programming language.

 

**Table 6. Decision rules.**

| 1 | (HemoglobinLevelgdL>=64)&(MCHCgdL>=40) =>(Class>=1) |
|---|---|
| 2 | (MCHCgdL>=37) &(RDWCV>=30) =>(Class>=1) |
| 3 | (RDWCV>=41) &(BTD>=6837209302)=>(Class>=1) |
| 4 | (HemoglobinLevelgdL>=51)&(RDWCV>=41)&(MI>=19452054790)&(BTD>=4843137255)=>(Class>=1) |
| 5 | (RDWCV>=40)&(MI>=29508196720)&(BTD>=4979591837)=>(Class>=1) |
| 6 | (RDWCV>=35)&(MI>=46428571430)&(BTD>=4600000000)=>(Class>=1) |
| 7 | (MCHCgdL>=30)&(RDWCV>=35)&(MI>=26739926740)&(BTD>=4475409836)=>(Class>=1) |
| 8 | (MCHCgdL>=33)&(RDWCV>=35)&(BTD>=4101694915)=>(Class>=1) |
| 9 | (RDWCV>=38)&(MI>=34297520660)&(BTD>=4101694915) =>(Class>=1) |
| 10 | (HemoglobinLevelgdL>=59)&(RDWCV>=35)&(MI>=17447916670)&(BTD>=4898305085)=>(Class>=1) |
| 11 | (HemoglobinLevelgdL>=45)&(RDWCV>=37)&(MI>=21455938700)&(BTD>=5800000000)=>(Class>=1) |
| 12 | (MCHCgdL>=30)&(MI>=119481000000)&(BTD>=3500000000)=>(Class>=1) |
| 13 | (MCHCgdL>=29)&(MI>=35185185190)&(BTD>=4500000000)=>(Class>=1) |
| 14 | (MCHCgdL>=33)&(RDWCV>=31)&(MI>=20000000000)&(BTD>=4320987654)=>(Class>=1) |
| 15 | (HemoglobinLevelgdL>=93)&(MI>=15200000000)&(BTD>=5376344086)=>(Class>=1) |
| 16 | (HemoglobinLevelgdL>=68)&(RDWCV>=30)&(MI>=19207317070)&(BTD>=4389830508)=>(Class>=1) |
| 17 | (HemoglobinLevelgdL>=56)&(RDWCV>=30)&(MI>=22950819670)&(BTD>=4389830508)=>(Class>=1) |
| 18 | (HemoglobinLevelgdL>=56)&(MI>=19207317070)&(BTD>=5857142857)=>(Class>=1) |
| 19 | (MCHCgdL>=27)&(RDWCV>=29)&(BTD>=7535211268)=>(Class>=1) |
| 20 | (HemoglobinLevelgdL>=117)&(MI>=10979729730)&(BTD>=4704545455)=>(Class>=1) |

## 6.1. Dominance based rough set approach

As discussed in the earlier section, we have collected the local dataset in coordination with the genetic testing lab and performed pre-processing to clean the data. In this research, we have used an imbalanced dataset with one class as a minority class (β-Thalassemia carrier, 17%). After pre-processing, we had 3416 records with 19 features. At this stage, we have applied heuristic rules based DRSA algorithm [22] to perform dimensionality reduction.

We have included a few indexes to check the classification accuracy of β-Thalassemia carrier detection, i.e., Menzer Index, Green and King index, Red Cell Distribution Width Index (RDWI), etc. These indexes successfully classify carriers, but we obtained better results when we incorporated a new index based on the red blood cell and hemoglobin ratio, beta thalassemia detector (BTD). At this stage, we once again employed DRSA algorithm [22] and finally obtained five features-based reduce set, called 'reducts' in DRSA. Based on the reduced feature set, we obtained lower approximation sets for complete data and then applied VC-Domlem algorithm to extract rules for classification [25]. We have trained the model on 80% data and performed testing on 20% data. The proposed model has achieved 91% accuracy on test data. The confusion matrix for the model shown in Fig 3 demonstrates strong classification performance, especially in identifying β-thalassemia carriers with minimal false negatives.

Detail results with confusion matrix PR and ROC curve are given in Table 7:

Figs 4 and 5 present the Precision-Recall and ROC curves for the DRSA model. With an AUC of 0.785, these curves confirm the model's ability to balance precision and recall in an imbalanced setting. To further evaluate the generalization of our proposed model, we have tested the model by giving 197 records (imbalance data) unseen and performing classification using these rules. The DRSA model maintained an impressive 89% accuracy. Fig 6 shows the confusion matrix on this data, indicating that the model generalizes well. Figs 7 and 8 display the PR and ROC curves for the unseen data, showing an improved AUC of 0.870—further validating the model's robustness. Table 8 summarizes the performance of the DRSA model on unseen validation data.

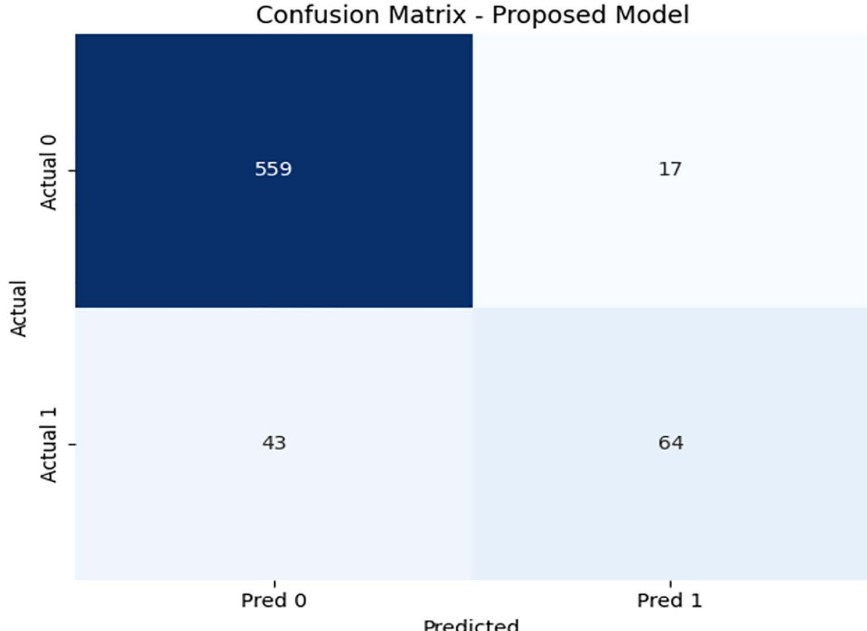

**Fig 3. Confusion matrix heatmap for the proposed DRSA-based model.**

**Table 7. Performance metrics of the proposed DRSA-based model.**

| Metric | Precision | Recall (Sensitivity) | F1-Score | Specificity | FPR | FNR | AUC | Support |
|---|---|---|---|---|---|---|---|---|
| **Class 0** | 0.93 | 0.97 | 0.95 | 0.60 | 0.40 | 0.03 | 0.785 | 576 |
| **Class 1** | 0.79 | 0.60 | 0.68 | 0.97 | 0.03 | 0.40 | 0.785 | 107 |
| **Accuracy** | – | – | 0.91 | – | – | – | – | 683 |
| **Macro Avg** | 0.86 | 0.78 | 0.81 | 0.79 | 0.22 | 0.22 | 0.785 | 683 |
| **Weighted Avg** | 0.91 | 0.91 | 0.91 | – | – | – | – | 683 |

## 6.2. Random Forest

To compare the performance of our proposed framework, we have implemented a random forest algorithm on imbalanced data and performed evaluation measures. We have obtained 90% accuracy on test data. Fig 9 presents the confusion matrix of the Random Forest model on the training data. While it performs well overall, its recall for the minority class remains lower than that of DRSA. Figs 10 and 11 depict the Random Forest PR and ROC curves, showing competitive AUC but reduced sensitivity for β-thalassemia carriers. Tables 9 and 10 presents the Random Forest model's performance on both the training and unseen datasets.

To further evaluate the generalization of the model, we test the random forest classifier on unseen data, consisting of 197 records. On unseen data, Fig 12 shows the confusion matrix, where accuracy dropped to 86%, indicating slight over-fitting. Figs 13 and 14 illustrates the PR & ROC curve on this unseen dataset, which, while respectable, confirms DRSA's superior performance.

## 6.3. Support Vector Machine (SVM) and other techniques

We have used SVM learning model to design a classifier. SVM is a widely used supervised learning algorithm for binary classification tasks [30] due to its simplicity in terms of parameter adjustment. However, in the case of a large dataset,

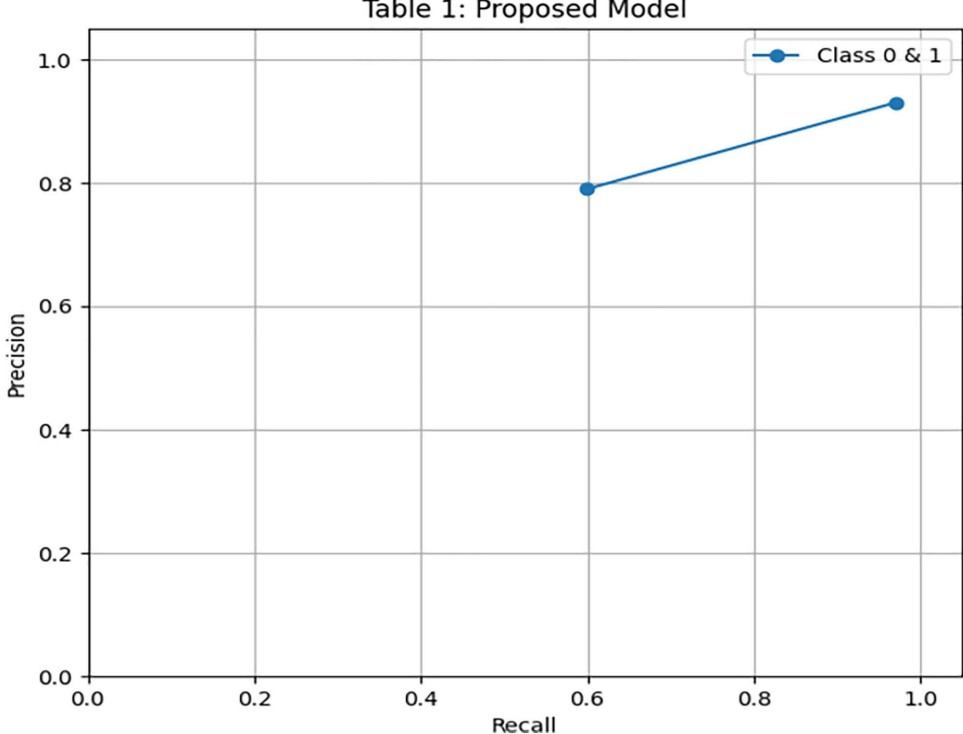

**Fig 4. PR & ROC curve showing the AUC performance of the proposed DRSA-based model.**

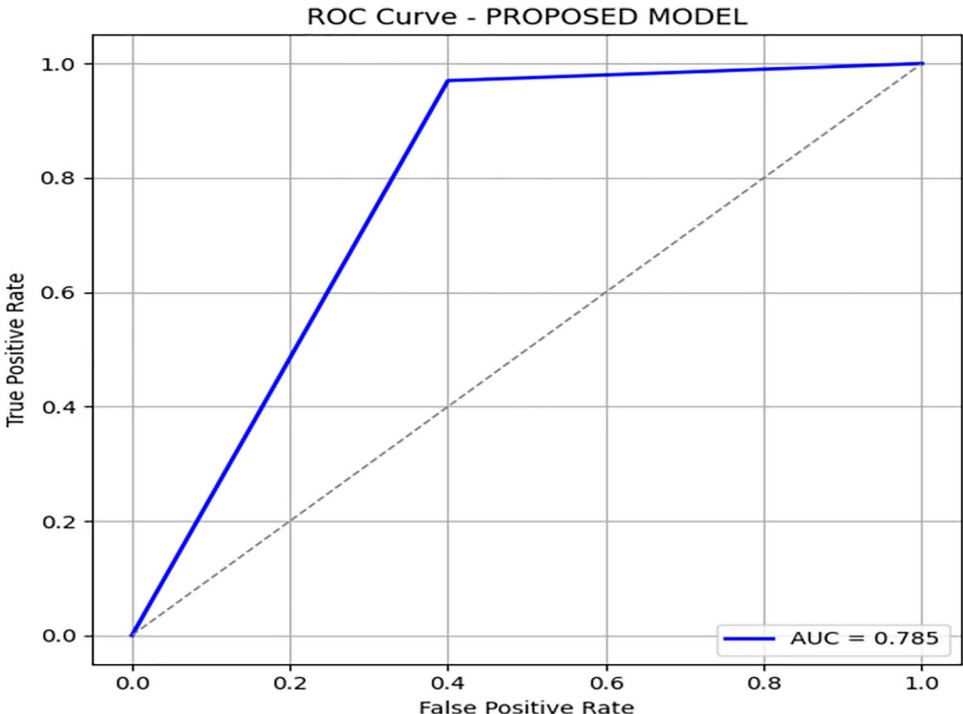

**Fig 5. PR & ROC curve showing the AUC performance of the proposed DRSA-based model.**

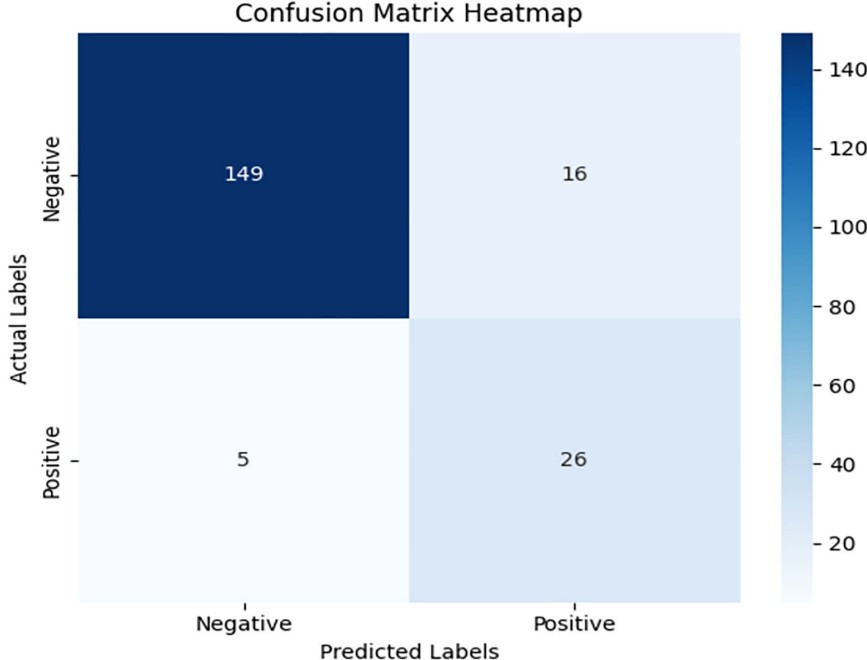

**Fig 6. Confusion matrix heatmap for the proposed DRSA-based model on Unseen Data.**

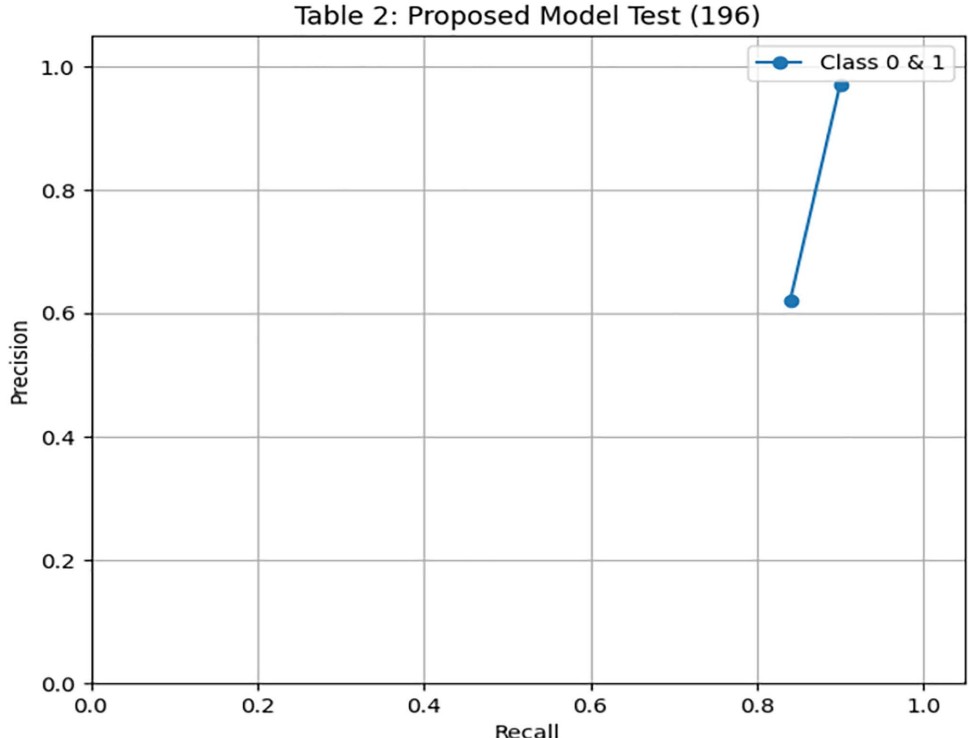

**Fig 7. PR & ROC curve showing the AUC performance of the proposed DRSA-based model on Unseen Data.**

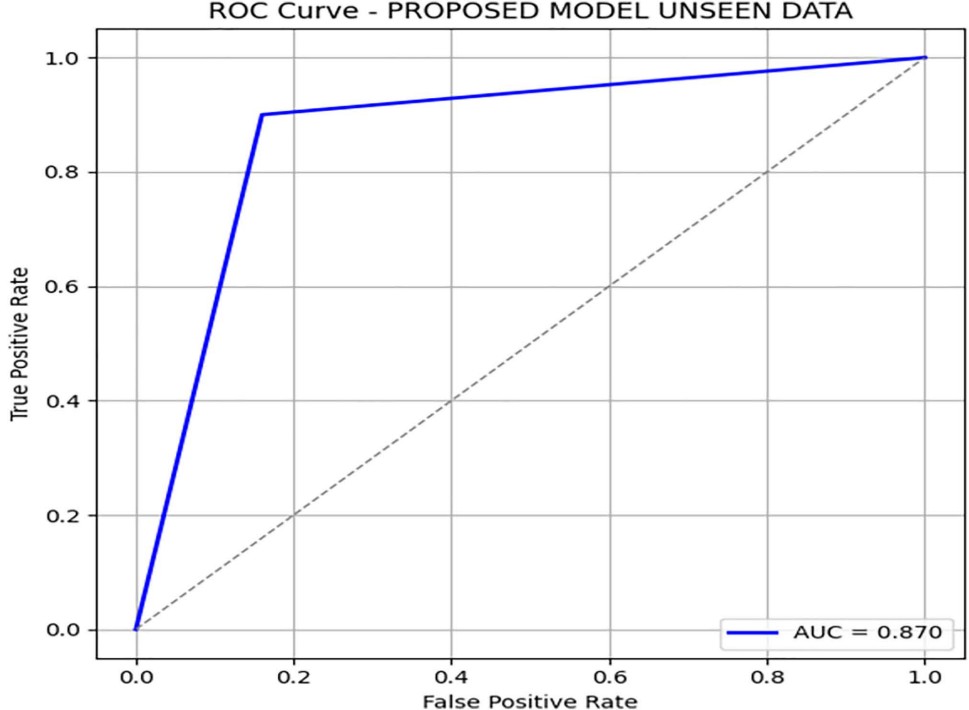

**Fig 8. PR & ROC curve showing the AUC performance of the proposed DRSA-based model on Unseen Data.**

**Table 8. Performance metrics of the proposed DRSA-based model on the Unseen data.**

| Metric | Precision | Recall (Sensitivity) | F1-Score | Specificity | FPR | FNR | AUC | Support |
|---|---|---|---|---|---|---|---|---|
| **Class 0** | 0.97 | 0.90 | 0.93 | 0.84 | 0.16 | 0.10 | 0.870 | 165 |
| **Class 1** | 0.62 | 0.84 | 0.71 | 0.90 | 0.10 | 0.16 | 0.870 | 31 |
| **Accuracy** | – | – | 0.89 | – | – | – | – | 196 |
| **Macro Avg** | 0.79 | 0.87 | 0.82 | 0.87 | 0.13 | 0.13 | 0.870 | 196 |
| **Weighted Avg** | 0.91 | 0.89 | 0.90 | – | – | – | – | 196 |

its application is limited due to high computational time. We have implemented SVM on 3416 records and obtained 86% accuracy. For SVM, we obtained the same accuracy 86%, when applied to a classifier on unseen data. In the case of an imbalanced dataset, we have found that our proposed model outperformed SVM in terms of accuracy and other evaluation parameters. Fig 15 illustrates the confusion matrix of the SVM model. It shows very high specificity for the majority class, but poor recall for the minority class. Figs 16 and 17 highlight SVM's underperformance with an AUC of just 0.56, indicating limited capability to detect β-thalassemia carriers in imbalanced data. Table 11 details the evaluation metrics for the Support Vector Machine model.

To further investigate the performance of our proposed model, we have explored neural network architecture and designed a recurrent neural network, a deep neural network. We have achieved an accuracy of 85% with an imbalanced dataset of 3416 records. Fig 18 illustrates the confusion matrix of the DNN model. Figs 19 and 20 show the PR and ROC curves for the DNN model. The model achieves an AUC of 0.80, comparable to DRSA, but with slightly lower recall. Table 12 displays the performance scores of the Deep Neural Network model.

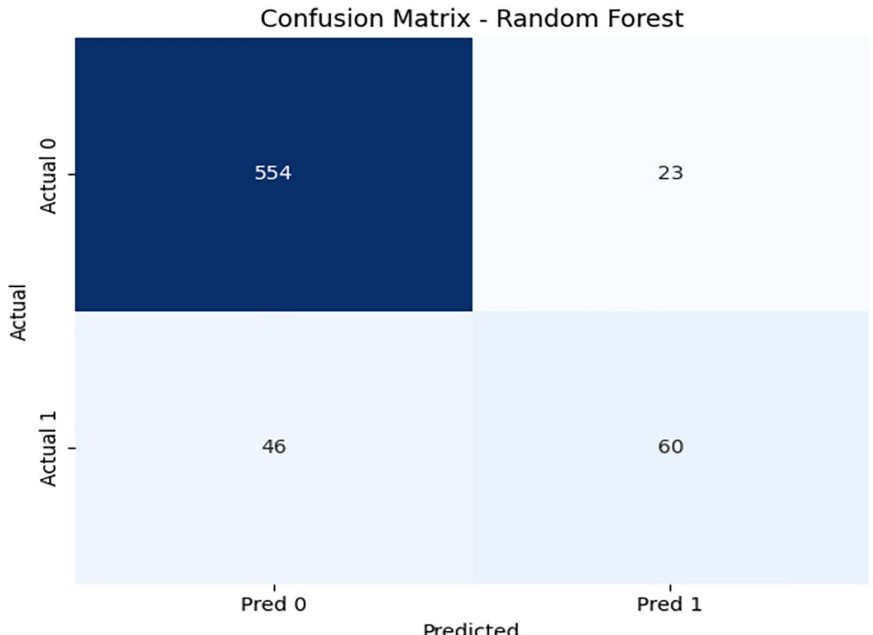

**Fig 9. Confusion matrix heatmap for the Random Forest model on the training dataset.**

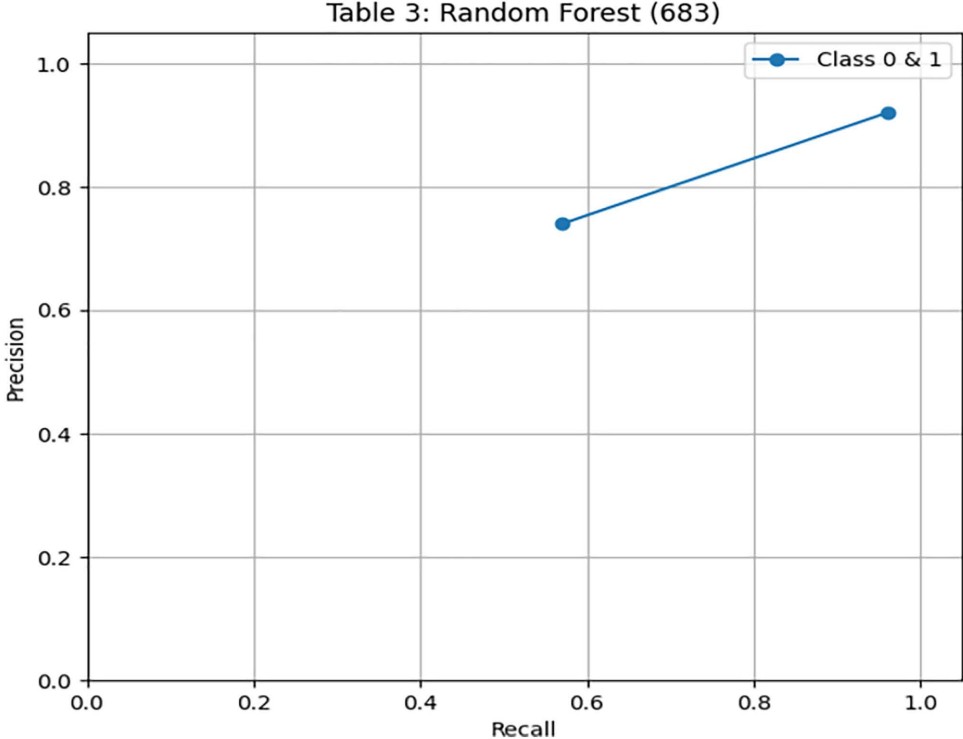

**Fig 10. PR & ROC curve showing the AUC performance of the Random Forest model on the training dataset.**

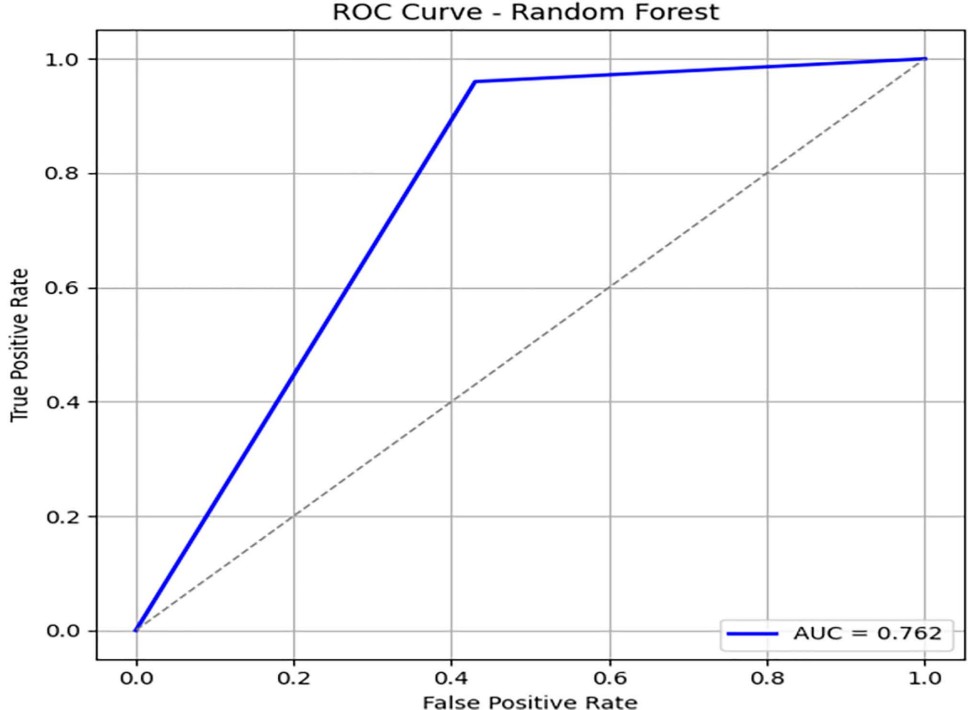

**Fig 11. PR & ROC curve showing the AUC performance of the Random Forest model on the training dataset.**

**Table 9. Performance metrics of the Random Forest model on the training dataset.**

| Metric | Precision | Recall (Sensitivity) | F1-Score | Specificity | FPR | FNR | AUC | Support |
|---|---|---|---|---|---|---|---|---|
| **Class 0** | 0.92 | 0.96 | 0.94 | 0.57 | 0.43 | 0.04 | 0.765 | 577 |
| **Class 1** | 0.74 | 0.57 | 0.64 | 0.95 | 0.05 | 0.43 | 0.760 | 106 |
| **Accuracy** | – | – | 0.90 | – | – | – | – | 683 |
| **Macro Avg** | 0.83 | 0.76 | 0.79 | 0.76 | 0.24 | 0.24 | 0.762 | 683 |
| **Weighted Avg** | 0.90 | 0.90 | 0.90 | – | – | – | – | 683 |

**Table 10. Performance metrics of the Random Forest model on Unseen data.**

| Metric | Precision | Recall (Sensitivity) | F1-Score | Specificity | FPR | FNR | AUC | Support |
|---|---|---|---|---|---|---|---|---|
| **Class 0** | 0.96 | 0.87 | 0.91 | 0.81 | 0.19 | 0.13 | 0.840 | 165 |
| **Class 1** | 0.54 | 0.81 | 0.65 | 0.87 | 0.13 | 0.19 | 0.840 | 31 |
| **Accuracy** | – | – | 0.86 | – | – | – | – | 196 |
| **Macro Avg** | 0.75 | 0.84 | 0.78 | 0.84 | 0.16 | 0.16 | 0.840 | 196 |
| **Weighted Avg** | 0.89 | 0.86 | 0.87 | – | – | – | – | 196 |

Fig 21 displays the RNN confusion matrix. Its precision and recall are modest, especially for the minority class. Figs 22 and 23 depict the PR and ROC curves for the RNN model, with an AUC of 0.725, reinforcing its weaker performance compared to DRSA. The evaluation metrics for the Recurrent Neural Network are shown in Table 13.

From these experiments, we have concluded that our proposed model outperformed ML models and neural network while classifying imbalance dataset as shown in Table 14 and Fig 24.

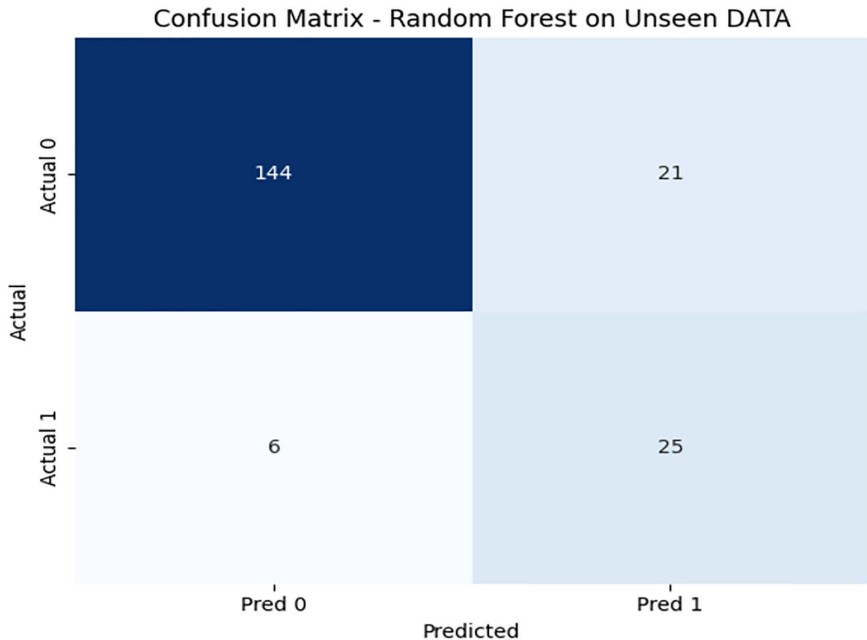

**Fig 12. Confusion matrix heat map for the Random Forest model on Unseen Data.**

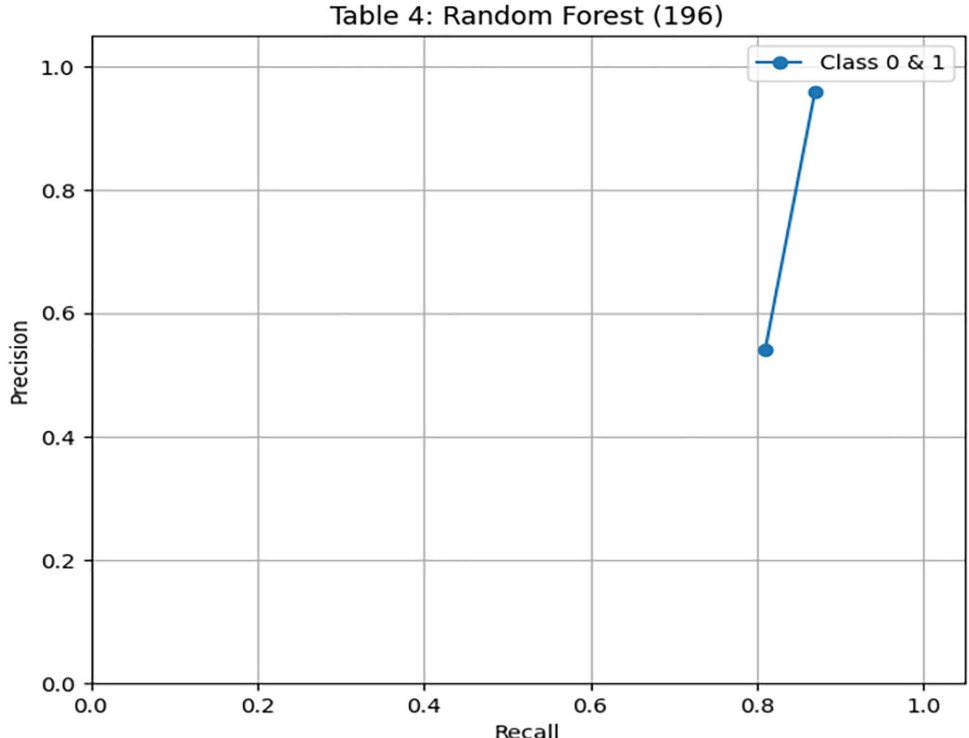

**Fig 13. ROC curve showing the AUC performance of the Random Forest model on Unseen Data.**

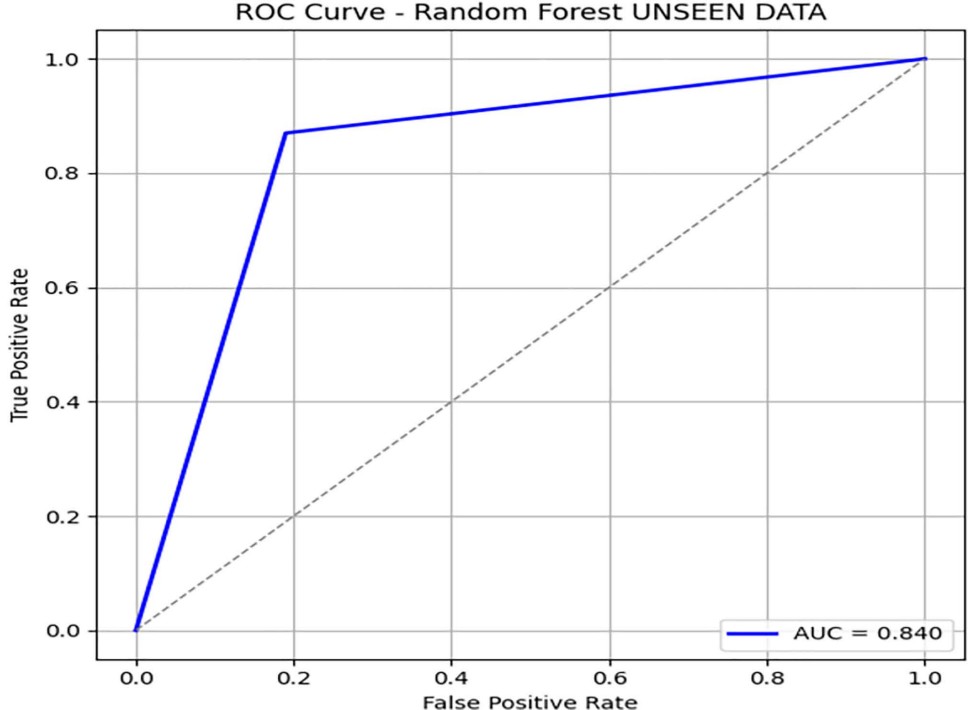

**Fig 14. ROC curve showing the AUC performance of the Random Forest model on Unseen Data.**

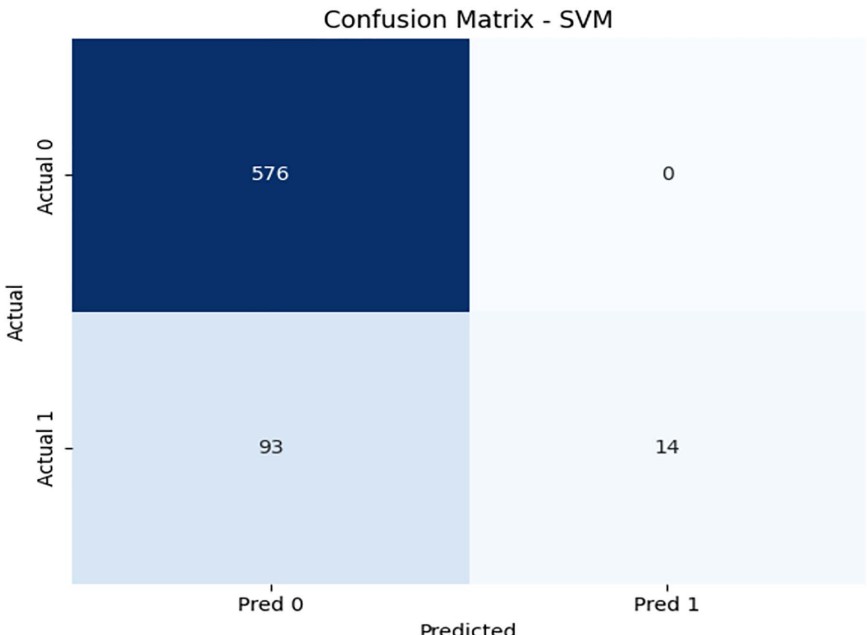

**Fig 15. Confusion matrix heatmap for the Support Vector Machine (SVM) model on training data.**

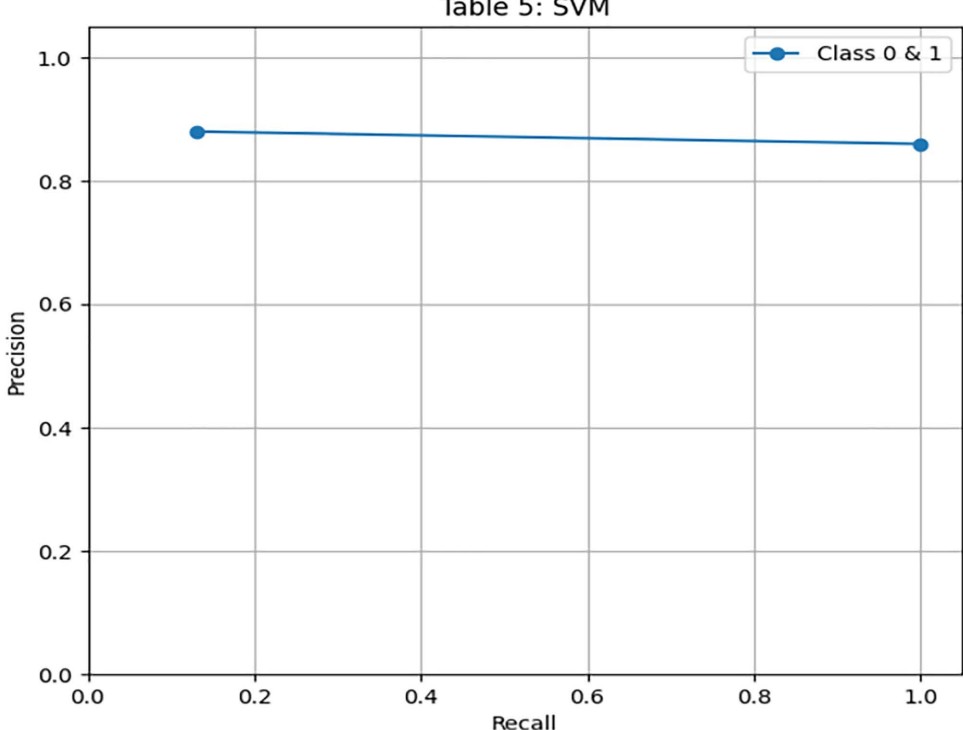

**Fig 16. PR & ROC curve showing the AUC performance of the Support Vector Machine (SVM) model on training Data.**

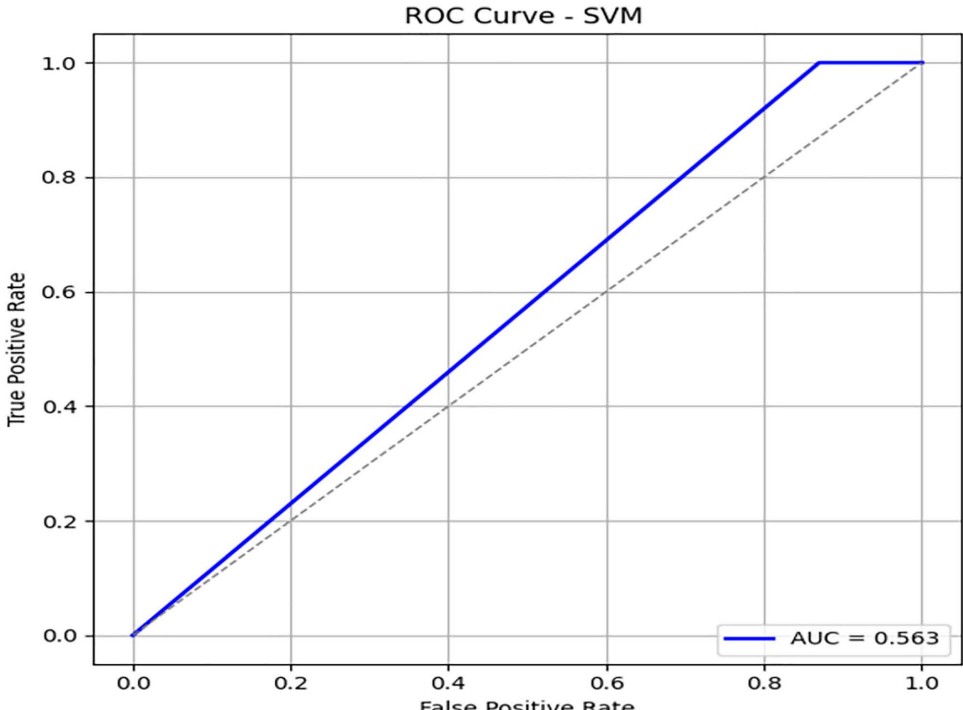

**Fig 17. PR & ROC curve showing the AUC performance of the Support Vector Machine (SVM) model on training Data.**

**Table 11. Performance metrics of the Support Vector Machine (SVM) model on training data.**

| Metric | Precision | Recall (Sensitivity) | F1-Score | Specificity | FPR | FNR | AUC | Support |
|---|---|---|---|---|---|---|---|---|
| **Class 0** | 0.86 | 1.00 | 0.92 | 0.13 | 0.87 | 0.00 | 0.565 | 576 |
| **Class 1** | 0.88 | 0.13 | 0.23 | 0.99 | 0.01 | 0.87 | 0.560 | 107 |
| **Accuracy** | – | – | 0.86 | – | – | – | – | 683 |
| **Macro Avg** | 0.87 | 0.56 | 0.58 | 0.56 | 0.44 | 0.44 | 0.563 | 683 |
| **Weighted Avg** | 0.86 | 0.86 | 0.81 | – | – | – | – | 683 |

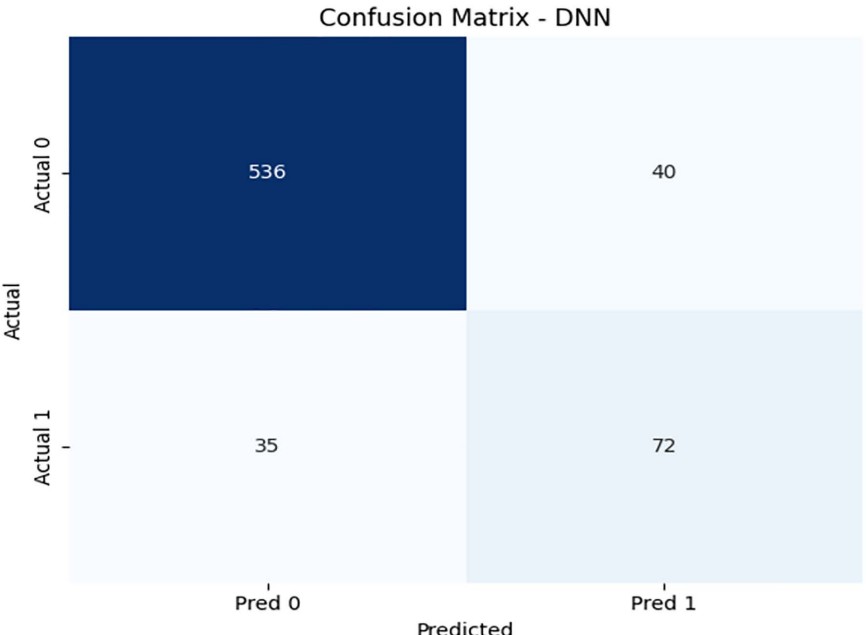

**Fig 18. Confusion matrix heatmap for the Deep Neural Network (DNN) model on training data.**

We want to clarify that the misclassifications (false positives and false negatives) observed in our study were primarily due to the limited availability of minority class instances (β-thalassemia carriers) in the dataset. This scarcity of minority class examples constrained the diversity of data available to the model, impacting predictive accuracy. A comparison of the proposed model with other techniques has been given in Table 14.

Fig 24 offers a visual comparison of model accuracies. The proposed DRSA model leads with 91% accuracy on imbalanced data. Further, in terms of generalization, the proposed model achieved effective accuracy, precision, and recall when experimented with unseen data, as compared to Random Forest and SVM, and comparable performance in comparison with deep neural network as shown in Table 15 and Fig 25. DRSA again achieves the highest accuracy, confirming its robustness and generalization capability.

### 6.4. Discussion

#### 6.4.1. Performance evaluation of proposed DRSA model.

- Recall (Sensitivity) for Class 1 (minority class) is 0.60 (683 samples) and 0.84 (196 samples), showing that DRSA performs particularly well in identifying true positive cases, especially in the smaller test set.

- Specificity for Class 1 is 0.97 (683) and 0.90 (196), meaning false positives are low—an important factor in reducing unnecessary treatments.

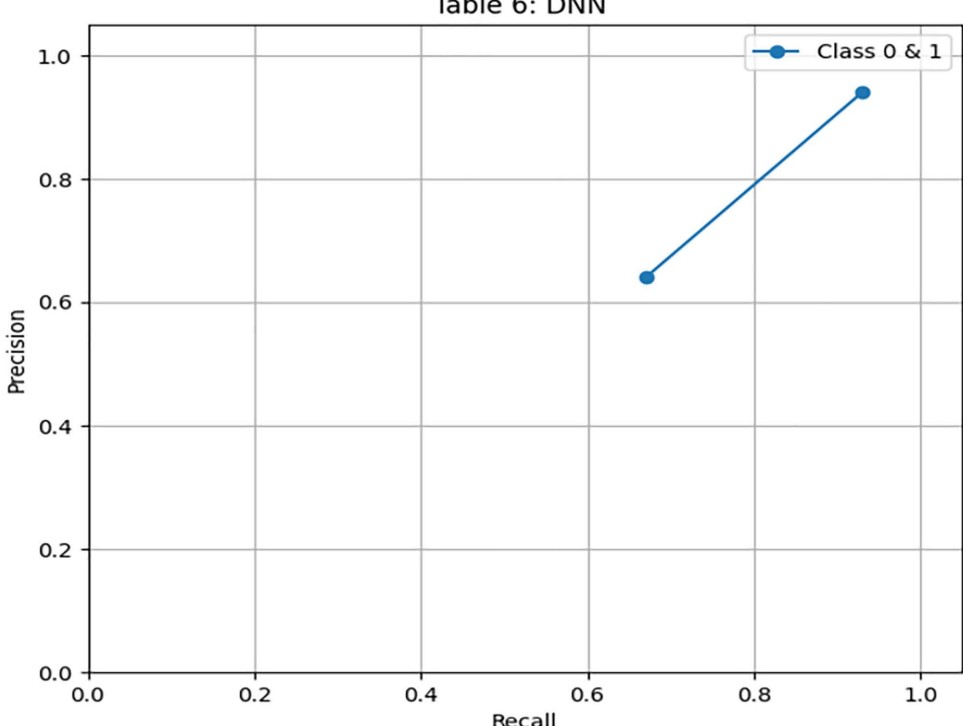

**Fig 19. PR & ROC curve showing the AUC performance of the Deep Neural Network (DNN) model on training data.**

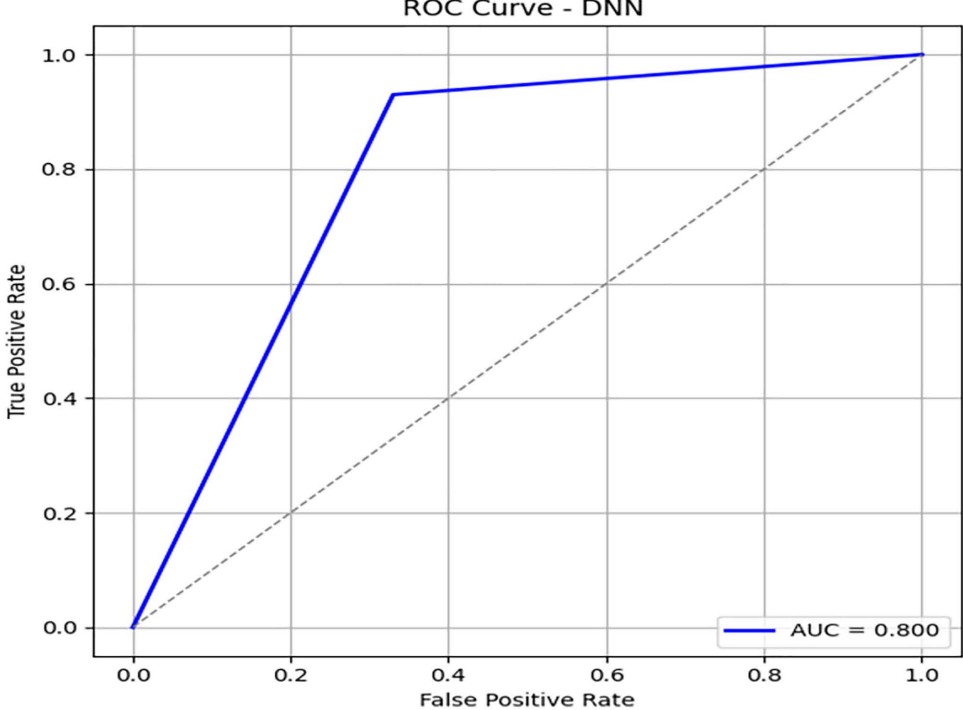

**Fig 20. PR & ROC curve showing the AUC performance of the Deep Neural Network (DNN) model on training data.**

**Table 12.  Performance metrics of the Deep Neural Network (DNN) model on training data.**

| Metric | Precision | Recall (Sensitivity) | F1-Score | Specificity | FPR | FNR | AUC | Support |
|---|---|---|---|---|---|---|---|---|
| **Class 0** | 0.94 | 0.93 | 0.93 | 0.67 | 0.33 | 0.07 | 0.80 | 576 |
| **Class 1** | 0.64 | 0.67 | 0.65 | 0.93 | 0.07 | 0.33 | 0.80 | 107 |
| **Accuracy** | – | – | 0.89 | – | – | – | – | 683 |
| **Macro Avg** | 0.79 | 0.80 | 0.79 | 0.80 | 0.20 | 0.20 | 0.80 | 683 |
| **Weighted Avg** | 0.89 | 0.89 | 0.89 | – | – | – | – | 683 |

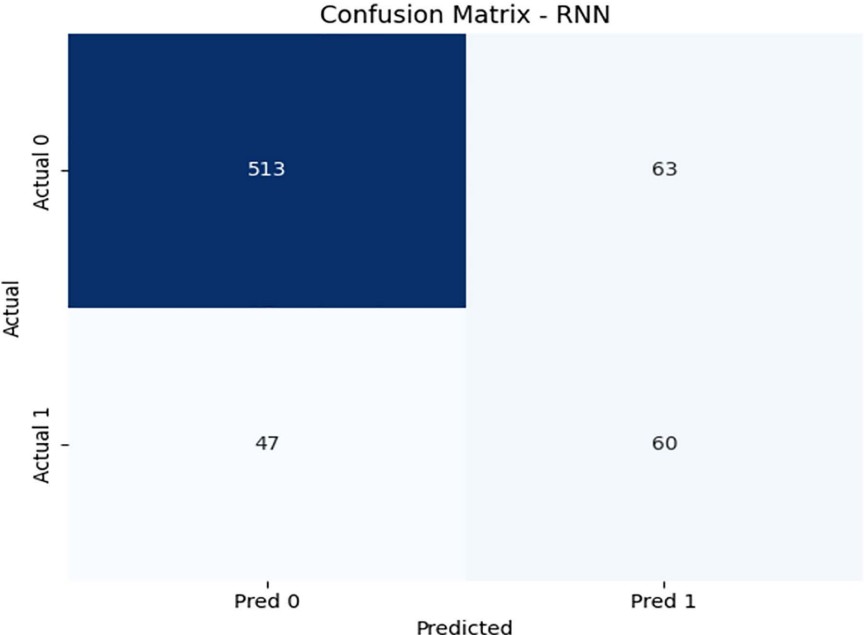

**Fig 21.  Confusion matrix heatmap for the Recurrent Neural Network (RNN) model on training data.**

- The AUC of 0.785 (683) and 0.870 (196) indicates a balanced and robust performance across both classes.

- DRSA maintains a high F1-score for Class 1 (0.68 and 0.71), reflecting a good balance between precision and recall.

DRSA demonstrates a strong capability to detect minority class cases while keeping false positives low. It shows generalizability across datasets (80/20 and holdout), outperforming others in recall and AUC—key metrics in medical diagnosis.

### 6.4.2.  Comparison with other models.

I.  **Random Forest**

- Slightly lower recall for Class 1 (0.57 and 0.81) compared to DRSA (0.60 and 0.84).

- AUC is lower: 0.760 (683) and 0.840 (196), compared to DRSA's 0.785/0.870.

- While Random Forest shows good precision for Class 0, it underperforms in detecting the minority class relative to DRSA.

Fig 24 compares the accuracy of all models under imbalanced conditions, clearly showing the superiority of the proposed DRSA model. Fig 26 presents the AUC values of each model. DRSA records the highest AUC, demonstrating excellent

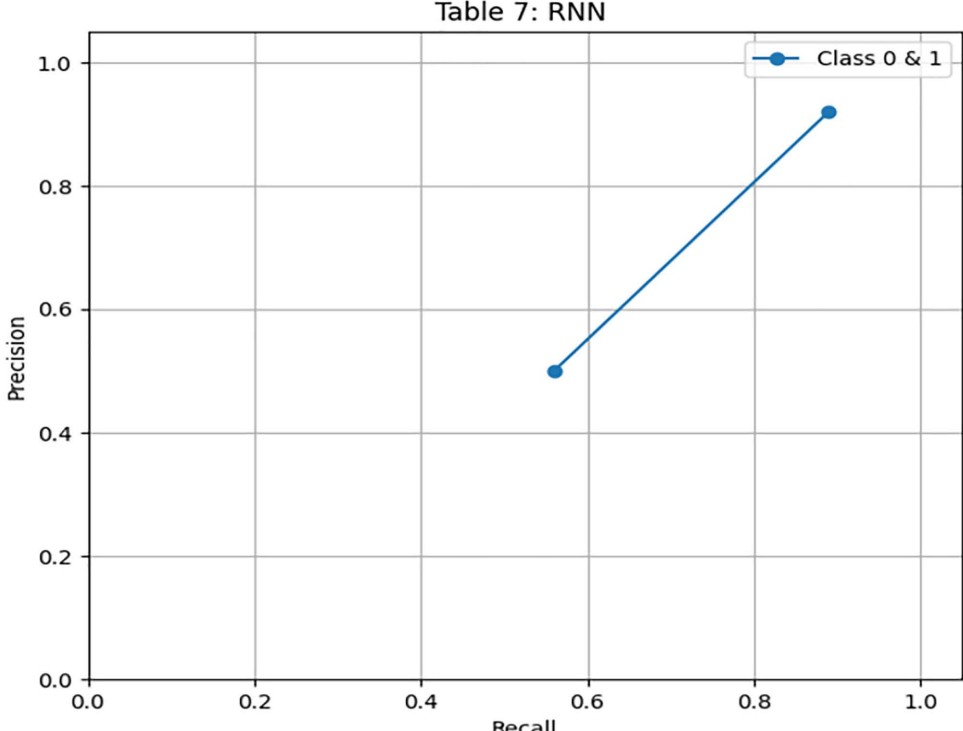

**Fig 22. PR & ROC curve showing the AUC performance of the Recurrent Neural Network (RNN) model on training data.**

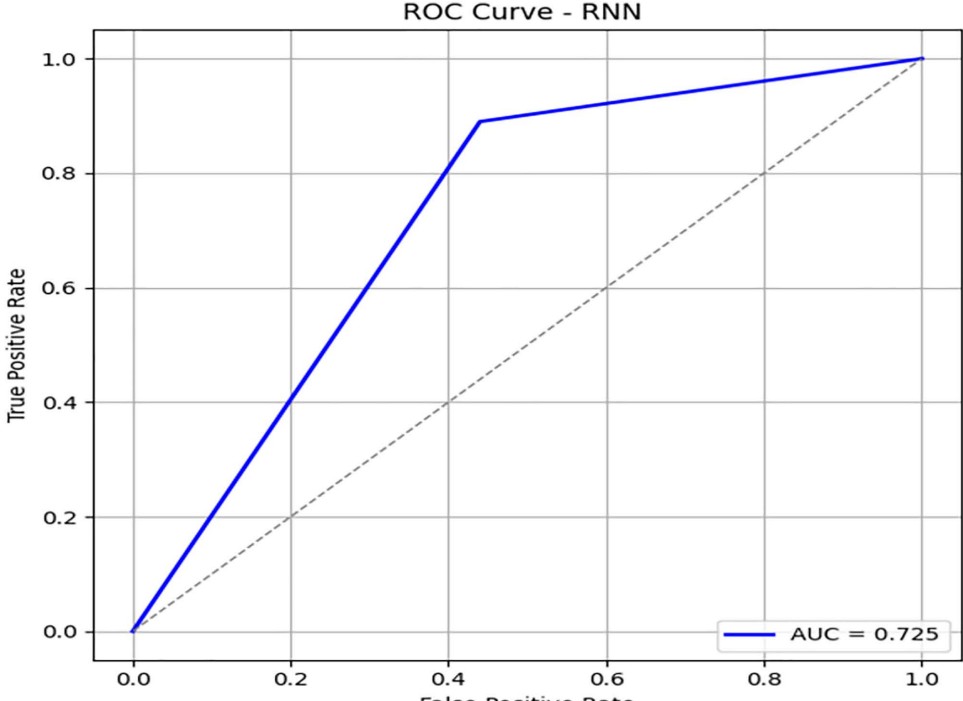

**Fig 23. PR & ROC curve showing the AUC performance of the Recurrent Neural Network (RNN) model on training data.**

**Table 13. Performance metrics of the Recurrent Neural Network (RNN) model on training data.**

| Metric | Precision | Recall (Sensitivity) | F1-Score | Specificity | FPR | FNR | AUC | Support |
|---|---|---|---|---|---|---|---|---|
| **Class 0** | 0.92 | 0.89 | 0.90 | 0.56 | 0.44 | 0.11 | 0.725 | 576 |
| **Class 1** | 0.50 | 0.56 | 0.53 | 0.89 | 0.11 | 0.44 | 0.725 | 107 |
| **Accuracy** | – | – | 0.84 | – | – | – | – | 683 |
| **Macro Avg** | 0.71 | 0.73 | 0.72 | 0.725 | 0.275 | 0.275 | 0.725 | 683 |
| **Weighted Avg** | 0.85 | 0.84 | 0.84 | – | – | – | – | 683 |

**Table 14. Comparison of proposed model.**

| Sr# | Model | Accuracy |
|---|---|---|
| 1 | Proposed Model | 91% |
| 2 | Random Forest | 90% |
| 3 | SVM | 86% |
| 4 | RNN | 84% |
| 5 | Deep Neural Network | 89% |

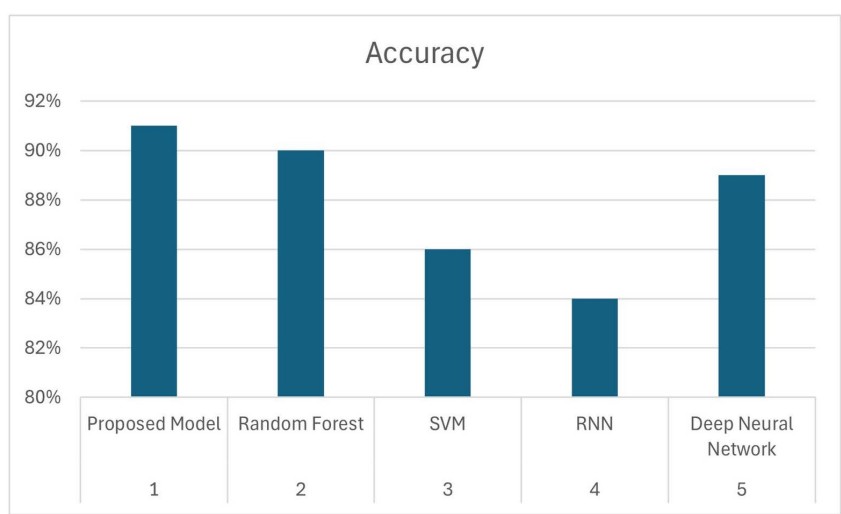

**Fig 24. Comparison of Proposed Model.**

**Table 15. Comparison of proposed model using unseen data.**

| Sr# | Model | Accuracy |
|---|---|---|
| 1 | Proposed Model | 89% |
| 2 | Random Forest | 86% |
| 3 | SVM | 86% |
| 4 | RNN | 85% |
| 5 | Deep Neural Network | 89% |

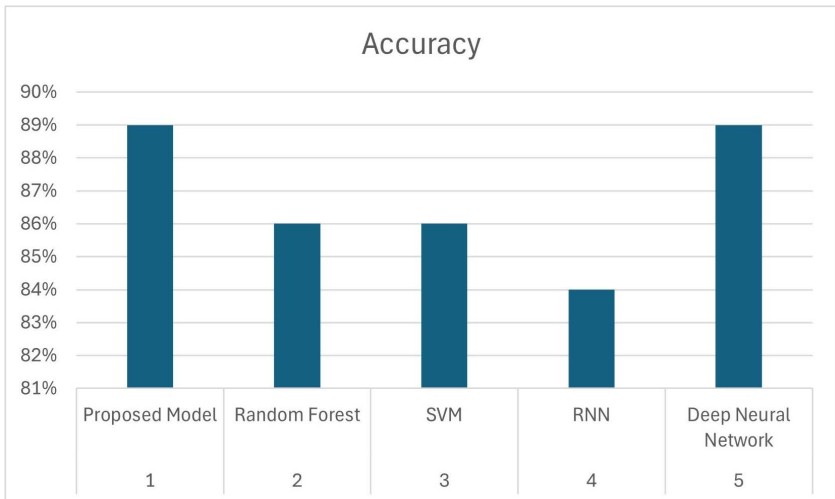

**Fig 25. Comparison of Proposed Model using Unseen Data.**

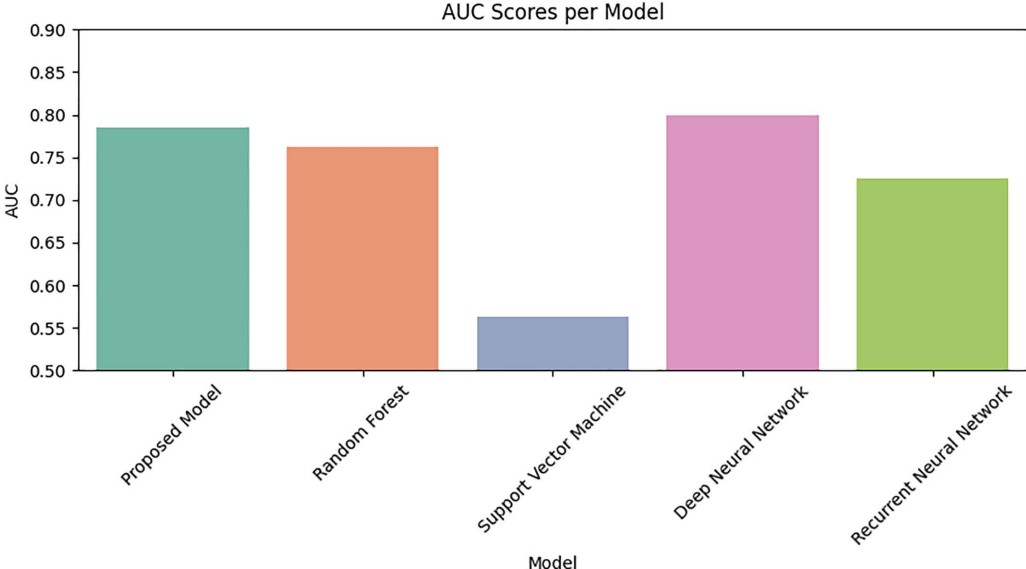

**Fig 26. Comparison of Proposed Model, RF, SVM, DNN, and RNN models using Area Under Curve on imbalanced datasets.**

overall classification capability. While Random Forest performs competitively, DRSA has a slightly better recall and AUC, making it more suitable for imbalanced clinical data.

II. **SVM**

- Recall for Class 1 is extremely low (0.13), indicating it fails to detect positive cases.

- Despite high precision, the F1-score for Class 1 is just 0.23, showing a lack of practical usefulness.

- The AUC is the lowest among all models: 0.56, indicating poor discrimination capability.

SVM fails under imbalanced conditions, highlighting the need for models like DRSA that better handle skewed distributions.

### III. DNN

Recall for Class 1: 0.67 — competitive with DRSA.

• Precision: 0.64, slightly lower than DRSA.

• AUC is 0.80, matching DRSA's best performance on the full dataset.

DNN performs well, but DRSA slightly outperforms it in F1 and precision, especially when considering generalization to unseen data.

### IV. RNN

• Recall for Class 1: 0.56, lower than DRSA.

• Precision: 0.50, also lower.

• AUC: 0.725, significantly below DRSA.

RNN is less effective in identifying minority class instances and has overall lower metric performance compared to DRSA. Fig 27 shows that Class 0 precision is comparable across models, but Fig 28 reveals DRSA's dominance in Class 1 precision (β-thalassemia carriers). Figs 29 and 30 analyze recall for Class 0 and Class 1 respectively, DRSA performs consistently and accurately, especially in minority class recall.

Lastly, Figs 31 and 32 display F1 scores for Class 0 and Class 1. The proposed DRSA model again outperforms alternatives, balancing precision and recall for both classes more effectively than any other tested model.

The Proposed Model consistently demonstrates higher sensitivity, balanced F1-score, and stronger AUC in both 80/20 split and holdout datasets. These advantages are particularly crucial in imbalanced medical datasets, where the ability to correctly identify minority class (e.g., patients with disease) is essential. The results support DRSA as a robust, clinically applicable method for decision support in healthcare settings.

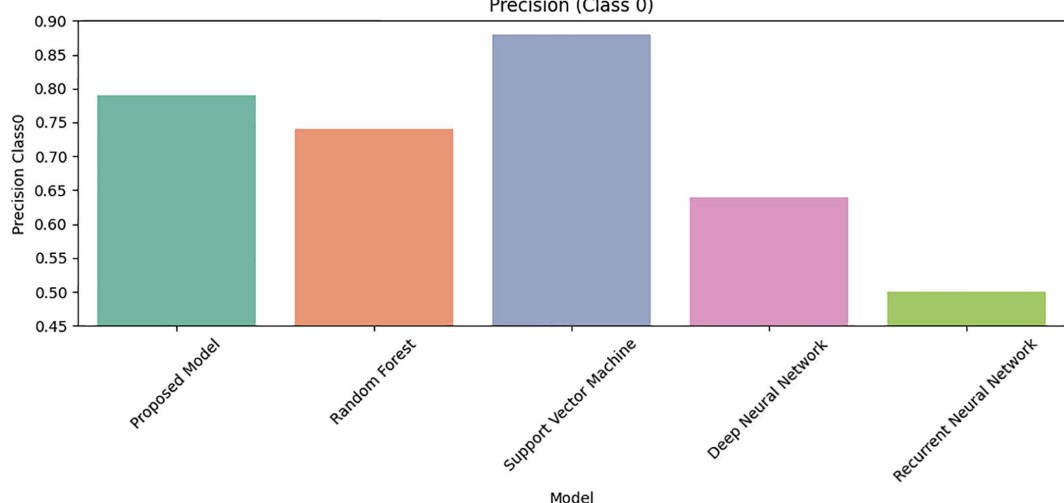

**Fig 27. Comparison of Proposed Model, RF, SVM, DNN, and RNN models using Precision for Class 0 on imbalanced datasets.**

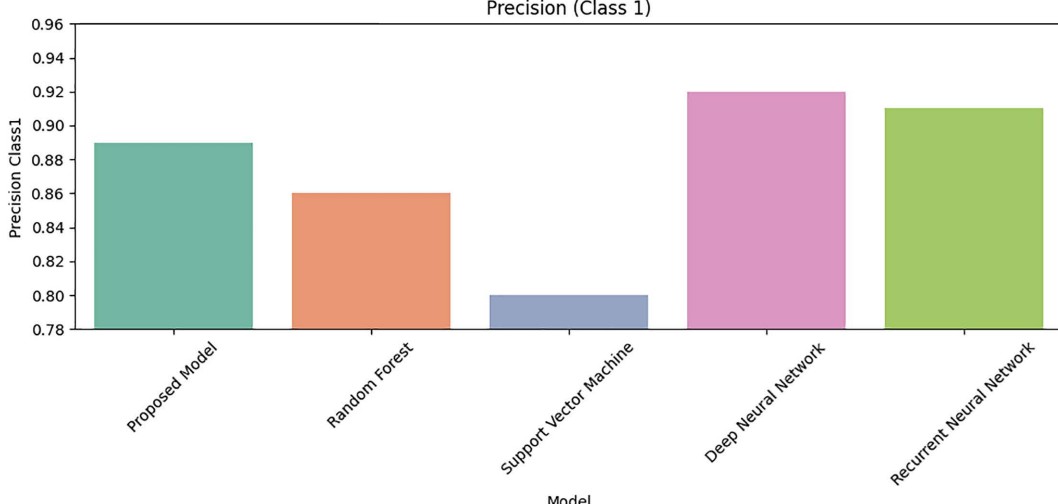

**Fig 28. Comparison of Proposed Model, RF, SVM, DNN, and RNN models using Precision for Class 1 on imbalanced datasets.**

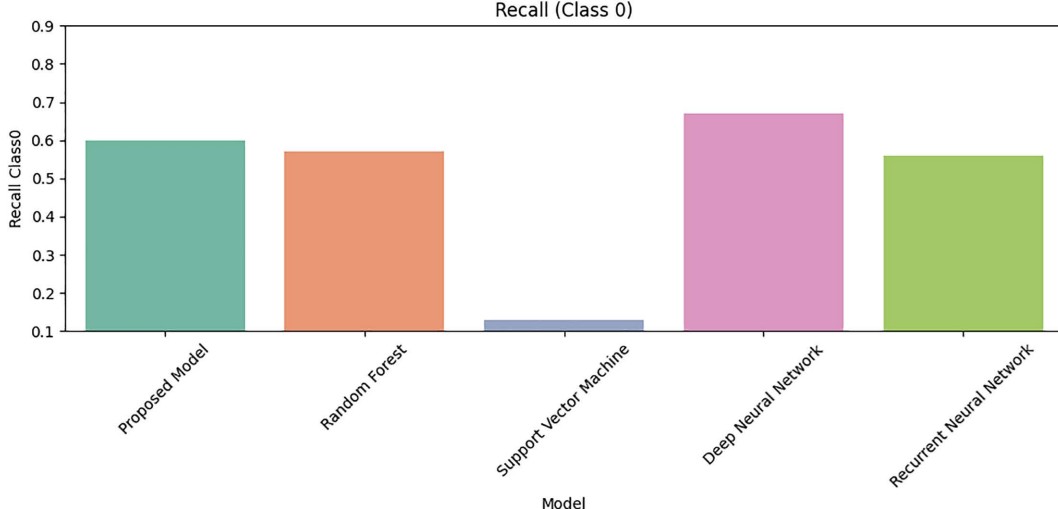

**Fig 29. Comparison of Proposed Model, RF, SVM, DNN, and RNN models using Recall for Class 0 on imbalanced datasets.**

**6.4.3. Statistical testing.** We have also performed statistical testing Paired T-Test to evaluate the performance of the proposed model. We have applied different models on same dataset therefore, we have used Paired T-Test.

I. **Comparison with Random Forest Algorithm**

$$\text{Proposed Model (DRSA) : Mean accuracy} = 90.83\% \ (\pm 0.94\%)$$

$$\text{Random Forest : Mean accuracy} = 89.97\% \ (\pm 0.80\%)$$

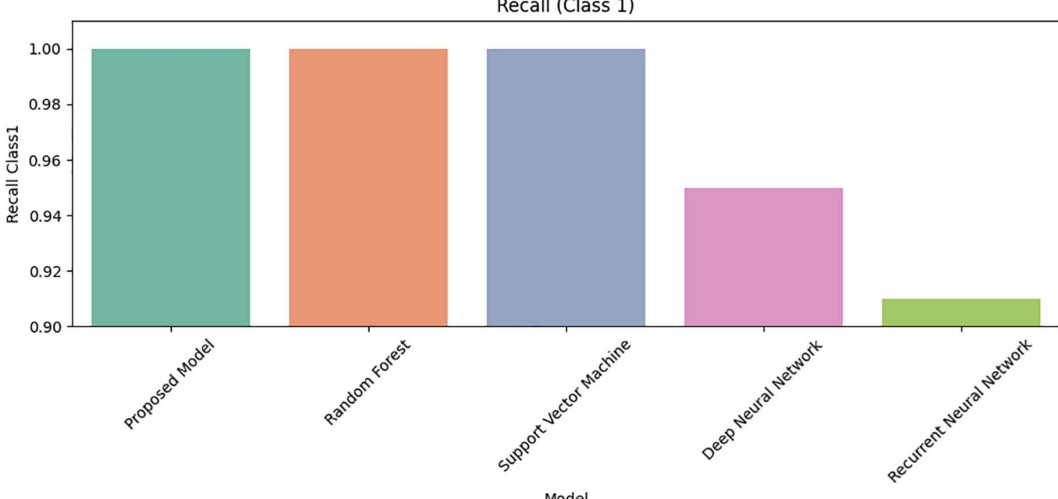

**Fig 30. Comparison of Proposed Model, RF, SVM, DNN, and RNN models using Recall for Class 1 on imbalanced datasets.**

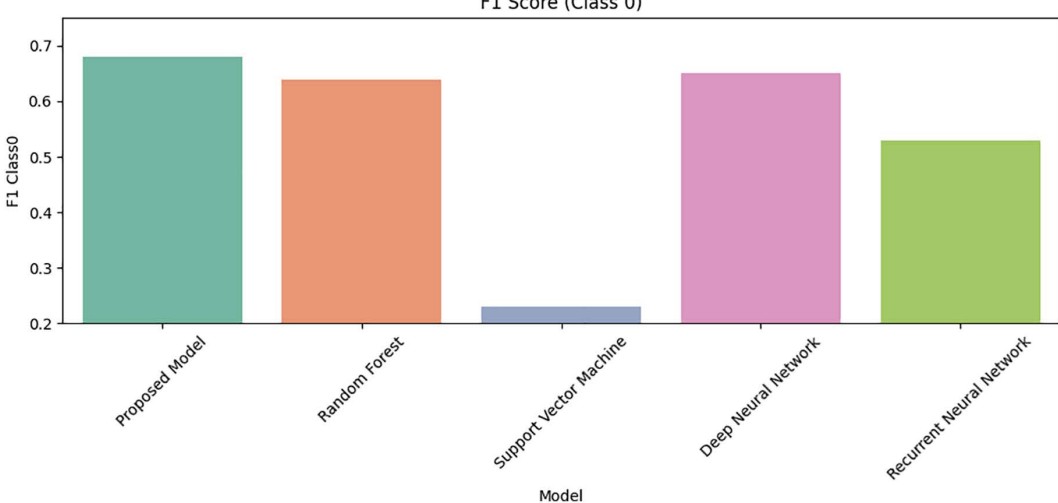

**Fig 31. Comparison of Proposed Model, RF, SVM, DNN, and RNN models using F1 Score for Class 0 on imbalanced datasets.**

The t-statistic is 2.60, with a corresponding p-value of approximately 0.017, which is below the conventional significance threshold (0.05). This indicates that the proposed model statistically significantly outperforms the Random Forest algorithm. Table 16 outlines the results of the paired t-test comparing the DRSA and Random Forest models.

These results emphasized that the proposed model provides a significant performance improvement over the Random Forest method.

## II. **Comparison with DNN**

The paired t-test results indicate a significant statistical difference between the proposedmodel and the Deep Neural Network (DNN) model:

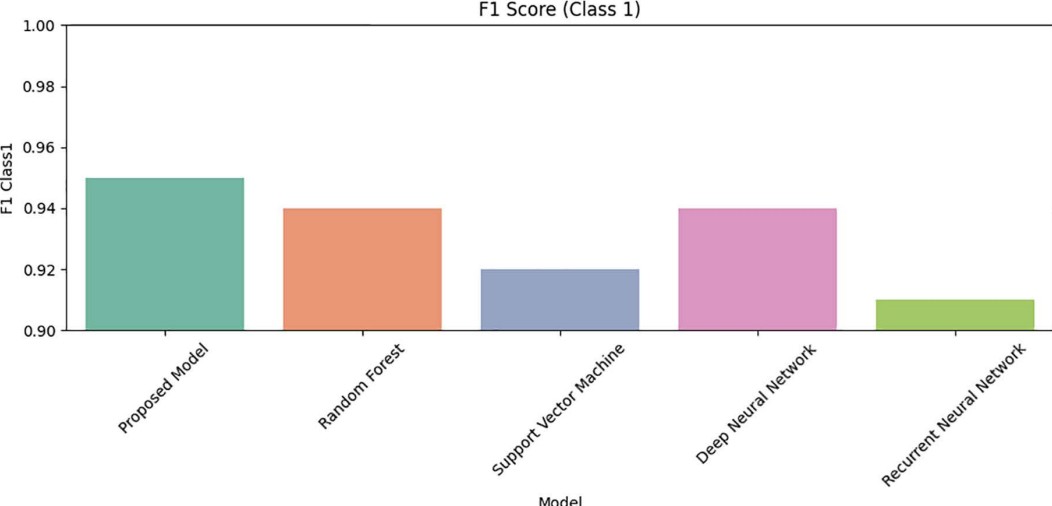

**Fig 32. Comparison of Proposed Model, RF, SVM, DNN, and RNN models using F1 Score for Class 1 on imbalanced datasets.**

**Table 16. T-test comparison of proposed model with Random Forest.**

| Classes/ Comparison | Model | Mean Accuracy | Standard Deviation | Metric | Value |
|---|---|---|---|---|---|
| 0 | Proposed Model | 0.908287 | 0.009357 | | |
| 1 | Random Forest | 0.899733 | 0.008001 | | |
| 0 | | | | t-statistic | 2.604742 |
| 1 | | | | p-value | 0.017408 |

**Table 17. T-test comparison of proposed model with DNN.**

| Classes/Comparison | Model | Mean Accuracy | Standard Deviation | Metric | Value |
|---|---|---|---|---|---|
| 0 | Proposed Model | 0.908287014 | 0.0093572 | | |
| 1 | Deep Neural Network | 0.887340249 | 0.009435274 | | |
| 0 | | | | t-statistic | 6.387048842 |
| 1 | | | | p-value | 3.99E-06 |

Proposed Model (DRSA) : Mean accuracy $=$ 90.83% ($\pm$0.94%)

Deep Neural Network : Mean accuracy $=$ 88.73% ($\pm$0.94%)

The calculated t-statistic is 6.39, and the corresponding p-value is approximately 0.000004, which is well below the common significance level (0.05). This means that the higher accuracy achieved by the proposed model compared to the DNN model is statistically significant. Table 17 displays the t-test results between DRSA and Deep Neural Network, confirming statistically significant superiority.

Therefore, it has been found that DRSA model not only matches but statistically outperforms the DNN approach with a significant margin, supported by the presented statistical analysis.

## 7. Conclusion and future work

In this research, we focused on local demographics due to the high percentage of β-thalassemia carrier in Pakistan. We have used data consisting of CBC features and assisted in the early identification of β-thalassemia carrier prior to genetic testing. Moreover, we considered an imbalanced dataset with 17% minority class (β-thalassemia carrier) and performed classification without using any normalization technique. We proposed a technique to identify high-risk patients by employing a dominance-based rough set approach and designing interpretable rules of classification. We have incorporated the Mentzer Index and proposed a new index to better classify carriers among non-carriers. To evaluate the performance of the proposed model, we implemented the proposed model and compared its performance with other algorithms when data is imbalanced. During this study, the proposed model proves very effective without using any technique to normalize the data. These results paved the way to further explore this field for more accurate and early diagnosis of beta-thalassemia carrier. Thus, specific interventions as well as personalized care strategies can be adopted to improve the lives of people.

In the future, we plan to refine our proposed model further to enhance accuracy and explore strategies aimed at reducing false positive and false negative predictions. Efforts will be made to expand the dataset to include a broader range of patient demographics, specifically considering age and sex, to improve the generalizability and robustness of the classification model.

Additionally, we will extend our research methodology to other hematological disorders, specifically targeting the identification and early detection of Sickle Cell anemia and Iron Deficiency anemia using CBC parameters. Furthermore, we intend to collaborate with Metropole Laboratories Private Limited (MPL) to develop a user-friendly front-end platform designed specifically for healthcare providers, facilitating seamless integration of our predictive model into clinical workflows and bridging the gap between advanced AI methodologies and practical clinical applications.

## Acknowledgments

The authors express their sincere gratitude to Metropole Laboratories, Islamabad, for their invaluable support in providing the Complete Blood Count (CBC) reports used in this study. Their collaboration and assistance were crucial in enabling the successful completion of this research.

During the preparation of this work the authors used Grammarly in order to improve readability. After using this tool/service, the authors reviewed and edited the content as needed and take full responsibility for the content of the publication.

## Author contributions

**Conceptualization:** Joddat Fatima.

**Formal analysis:** Joddat Fatima, Nadia Sultan.

**Investigation:** Faryal Nosheen.

**Methodology:** Saim Chishti, Faryal Nosheen.

**Software:** Saim Chishti, Faryal Nosheen.

**Supervision:** Faryal Nosheen, Joddat Fatima, Madiha Khalid.

**Validation:** Madiha Khalid.

**Writing – original draft:** Nadia Sultan.

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
