## [Decision Letter · Decision Letter 0]

15 Jul 2025

Dear Dr. Sultan,

Thank you for submitting your manuscript to PLOS ONE. After careful consideration, we feel that it has merit but does not fully meet PLOS ONE’s publication criteria as it currently stands. Therefore, we invite you to submit a revised version of the manuscript that addresses the points raised during the review process.

We look forward to receiving your revised manuscript.

Kind regards,

Nejat Mahdieh

Academic Editor

PLOS ONE

Journal Requirements:

4. In the online submission form, you indicated that your data will be submitted to a repository upon acceptance. We strongly recommend all authors deposit their data before acceptance, as the process can be lengthy and hold up publication timelines. Please note that, though access restrictions are acceptable now, your entire minimal dataset will need to be made freely accessible if your manuscript is accepted for publication. This policy applies to all data except where public deposition would breach compliance with the protocol approved by your research ethics board. If you are unable to adhere to our open data policy, please kindly revise your statement to explain your reasoning and we will seek the editor's input on an exemption.

Reviewers' comments:

Reviewer's Responses to Questions

**Comments to the Author**

1. Is the manuscript technically sound, and do the data support the conclusions?

Reviewer #1: Yes

Reviewer #2: Yes

2. Has the statistical analysis been performed appropriately and rigorously?

Reviewer #1: Yes

Reviewer #2: Yes

3. Have the authors made all data underlying the findings in their manuscript fully available?

Reviewer #1: Yes

Reviewer #2: No

4. Is the manuscript presented in an intelligible fashion and written in standard English?

Reviewer #1: Yes

Reviewer #2: Yes

Reviewer #1: The article entitled "From CBC to Clarity: Interpretable Detection of BetaThalassemia Carriers in Imbalanced Datasets: explains using Machine learing approaches to detect carriers of B thalassemia.

suggestion:

Hematologist could indicate the carriers by blood indexes, the difficulty is for boarder line indexes, if this method only detects these carriers please explain how it improves the detection rate.

Reviewer #2: This is a very interesting scientific study. The proposed approach is original, particularly in its use of the Dominance-based Rough Set Approach (DRSA) combined with the VC-Domlem algorithm to detect beta-thalassemia carriers using simple Complete Blood Count (CBC) data. The study is well-structured, methodologically sound, and shows strong potential for clinical application.

The manuscript is written in clear and well-structured English, and the figures are generally well presented. However, it would be helpful to include direct links to the cited references to facilitate access for readers. Additionally, the introduction should be made more concise.

I would also like to raise a question regarding the model's performance: the sensitivity for detecting carriers appears to be around 60%. What strategies do you propose to improve this sensitivity while maintaining the interpretability of the model?

**Do you want your identity to be public for this peer review?** For information about this choice, including consent withdrawal, please see our Privacy Policy

Reviewer #1: No

Reviewer #2: **Yes: ** Ibtihal ABOUCHABAKA

---

## [Author Response · Author response to Decision Letter 1]

31 Jul 2025

Thank you for your insightful suggestion regarding the ability of hematologists to identify β-thalassemia carriers using blood indices, and the particular challenge presented by borderline cases. To help you reconnect with your suggestions, we have reproduced your comments and uploaded in the attached file section.

---

## [Decision Letter · Decision Letter 1]

24 Aug 2025

From CBC to Clarity: Interpretable Detection of Beta-Thalassemia Carriers in Imbalanced Datasets

PONE-D-25-27555R1

Dear Dr. Sultan,

We’re pleased to inform you that your manuscript has been judged scientifically suitable for publication and will be formally accepted for publication once it meets all outstanding technical requirements.

Kind regards,

Nejat Mahdieh

Academic Editor

PLOS ONE

Additional Editor Comments (optional):

Reviewers' comments:

Reviewer's Responses to Questions

**Comments to the Author**

Reviewer #2: All comments have been addressed

2. Is the manuscript technically sound, and do the data support the conclusions?

Reviewer #2: Yes

3. Has the statistical analysis been performed appropriately and rigorously?

Reviewer #2: Yes

4. Have the authors made all data underlying the findings in their manuscript fully available?

Reviewer #2: Yes

5. Is the manuscript presented in an intelligible fashion and written in standard English?

Reviewer #2: Yes

Reviewer #2: Thank you for taking all of my comments into consideration when revising your manuscript. The work presented is of good quality. I congratulate you on this revised version and wish you all the best for the future.

However, I suggest that you take care to strengthen the statistical rigor and simplify certain formulations in future submissions to further improve the clarity and flow of the text.

**Do you want your identity to be public for this peer review?** For information about this choice, including consent withdrawal, please see our Privacy Policy

Reviewer #2: **Yes: ** Ibtihal Abouchabaka

---

## [Editor Report · Acceptance letter]

PONE-D-25-27555R1

PLOS ONE

Dear Dr. Sultan,

I'm pleased to inform you that your manuscript has been deemed suitable for publication in PLOS ONE. Congratulations! Your manuscript is now being handed over to our production team.

Kind regards,

on behalf of

Dr. Nejat Mahdieh

Academic Editor

PLOS ONE